# EPISTEMIC-AWARE VISION–LANGUAGE FOUNDATION MODEL FOR FETAL ULTRASOUND INTERPRETATION

## ABSTRACT

Recent medical vision-language models have shown promise on tasks such as VQA, report generation, and anomaly detection. However, most are adapted to structured adult imaging and underperform in fetal ultrasound, which poses challenges of multi-view image reasoning, numerous diseases, and image diversity. To bridge this gap, we introduce `FetalMind`, a medical AI system tailored to fetal ultrasound for both report generation and diagnosis. Guided by clinical workflow, we propose *Salient Epistemic Disentanglement* (SED), which injects an expert-curated bipartite graph into the model to decouple view-disease associations and to steer preference selection along clinically faithful steps via reinforcement learning. This design mitigates variability across diseases and heterogeneity across views, reducing learning bottlenecks while aligning the model's inference with obstetric practice. To train `FetalMind` at scale, we curate `FetalSigma-1M` dataset, the first large-scale fetal ultrasound report corpus, comprising 20K reports from twelve medical centers, addressing the scarcity of domain data. Extensive experiments show that `FetalMind` outperforms open- and closed-source baselines across all gestational stages, achieving +14% average gains and +61.2% higher accuracy on critical conditions while remaining efficient, stable, and scalable.

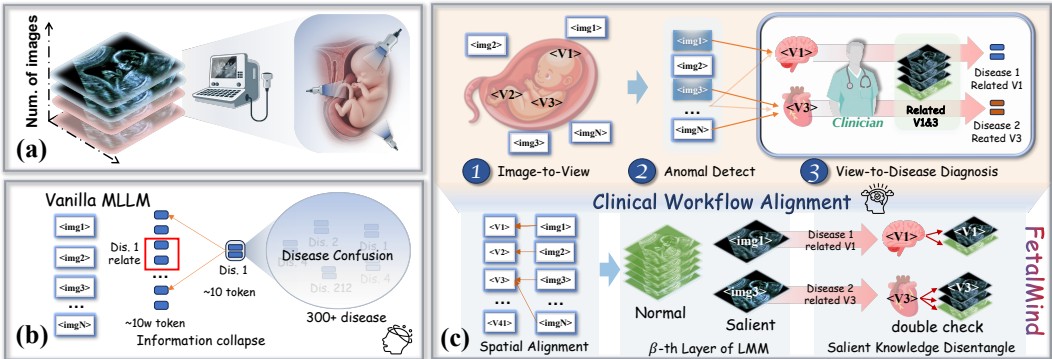

**Figure 1:** (a) Fetal ultrasound workflow. (b) Limitations of vanilla MLLMs on multi-view scans: ❶ A severe imbalance, with abundant visual tokens but limited textual supervision, induces representation collapse; ❷ Fetal imaging spans > 300 fine-grained diseases, markedly complicating robust diagnosis. (c) `FetalMind` aligns with the clinical workflow: view examination, abnormality detection, and disease tracing via knowledge.

## 1 INTRODUCTION

Ultrasound is the preferred tool for prenatal assessment, routinely used to track fetal growth, monitor pregnancy progression, and support clinical diagnosis (Salomon et al., 2022; Neilson et al., 1996). In contrast to adult imaging, fetal ultrasound requires integrating information across multiple views and gestational stages (Azad et al., 2024). Effective diagnosis must jointly consider developmental trajectories and early indicators of potential abnormalities (Lee et al., 2023). As illustrated in Figure 1a, fetal ultrasound typically involves many images with inconsistent view counts, substantial inter-case heterogeneity, and pronounced disease variability (Krishna & Kokil, 2024).

With the rise of deep learning, prior satisfactory works has decomposed fetal ultrasound into subtasks, e.g., biometric measurement, view classification, gestational age estimation, and anomaly

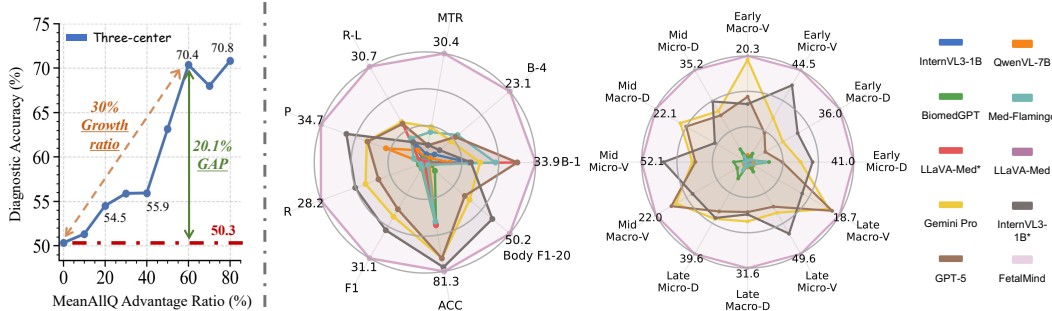

**Figure 2:** *Left:* Positive correlation ($> 0.3$) between diagnostic accuracy and the relative attention advantage of disease-related over non-disease views. Attention is measured by MeanALLQ, defined as the mean attention weight over all query tokens across layers and heads, and results are shown for Qwen-VL 2.5. *Right:* Multi-center evaluation of report generation and diagnosis with trimester-level diagnostic performance comparison.

analysis, achieving encouraging task-specific results (Fiorentino et al., 2023). More recently, several outstanding medical MLLM models have been proposed to handle cross-modal medical image and text instruction tasks, demonstrating significant results in experiments (Moor et al., 2023a).

However, when aligning multiple images with text, existing medical MLLM exhibit two critical issue (see Figure 1b): ❶ **Information collapse.** During disease-image alignment, diagnosis often contain only $\sim 10$ text tokens, while the associated image evidence may expand to $\sim 10^4$ visual tokens across views; the severe imbalance causes salient cues to be drowned out or ignored. ❷ **Disease confusion.** Fetuses present with multiple coexisting conditions, and disease-relevant views frequently overlap or partially align across slices. Such complexity hinders the inter-disease discriminability and results in confounded anomaly recognition and diagnosis. Consequently, reliable fetal ultrasound report generation and diagnosis remain unachieved with current deep learning approaches, limiting both clinical automation and decision support (Slimani et al., 2023).

The core challenge arises from the limitations of current MLLM approaches, which remain constrained to single-image, image–text alignment and therefore fail to capture anatomical development and latent abnormality associations across multiple views (Cheng et al., 2025; Liu et al.). In clinical practice, however, fetal ultrasound diagnosis does not rely on isolated images; it integrates spatial continuity and the developmental logic of anatomy across views (Carvalho et al., 2023). Existing models, lacking the ability to disentangle complementary information across views, often blur the correspondence between views and disease features (Arnaout et al., 2021). As illustrated in Figure 2 left, insufficient attention to disease-relevant views frequently leads to hallucinated or biased diagnoses, undermining reliability and diverging from established clinical workflows. In contrast, as illustrated in Figure 1c, obstetricians begin with a comprehensive survey of all views and progressively refine their focus on multiple views of specific regions to ensure thorough assessment.

Motivated by clinical workflows, we introduce *Spatial Alignment* to capture image-to-view correspondences and integrate it with *Salient Epistemic Disentanglement* through view preference optimization (SVPO). This synergy enhances the model's sensitivity to disease-bearing planes while explicitly injecting disease–plane associations, enabling the joint disentanglement of salient versus normal planes at both the case and view levels. Such modeling mirrors the reasoning process of obstetricians (Figure 1c), steering inference toward clinically grounded, auditable, and verifiable reports, thereby avoiding "isolated image → conclusion" shortcuts. To train `FetalMind` effectively, we construct the first large-scale fetal ultrasound report dataset, `FetalSigma-1M`. The dataset consists of real-world clinical data collected from 12 medical centers, covering 20,566 patients with 1.19M ultrasound images paired with expert-verified reports and diagnoses across early, mid, and late trimesters. As shown in Figure 2 right, `FetalMind` surpasses state-of-the-art medical MLLMs and general-purpose MLLMs (e.g., GPT-5) across multiple downstream tasks, highlighting its robustness and clinical applicability. To summarize, our contributions as follows:

❶ To the best of our knowledge, we present `FetalMind`, the first model for fetal ultrasound report generation and diagnosis capable of handling a variable number of views, with 1B and 7B versions. The model integrates salient epistemic disentanglement with salient view preference optimization and bipartite knowledge graph to capture disease–view associations, explicitly decouple salient from normal views at both the disease and view levels.

❷ We construct `FetalSigma-1M`, a large-scale multi-center benchmark comprising 1M multi-view ultrasound images and 20K paired clinical reports. The dataset spans all trimesters, covers all standard views, and includes over 300 diseases categories derived from real clinical examinations.

❸ We conduct extensive experiments showing that `FetalMind` achieves a 14% improvement in multi-center and zero-shot multi-device diagnosis, while maintaining strong robustness and generalization across diverse real-world clinical scenarios.

## 2 RELATED WORK

**Medical Multimodal Large Language Models.** Building on the success of general multimodal large language models (MLLMs) such as CLIP (Radford et al., 2021) and GPT-4 (Achiam et al., 2023), recent efforts have explored foundation models for medicine that learn unified image–text representations. LLaVA-Med augments biomedical imagery with open-ended dialogue and QA via large-scale chart–caption data and GPT-4–based instruction synthesis (Li et al., 2023). Med-PaLM accommodates text, images, and genomics under a single parameterization (Singhal et al., 2025). Several medical MLLMs also incorporate ultrasound data. BiomedGPT is an open, lightweight medical VLM supporting images, text, and tables (Zhang et al., 2024). HealthGPT unifies multimodal understanding and generation in an autoregressive framework (Lin et al., 2025). MedRegA provides a bilingual, general-purpose medical AI across eight modalities for both image- and region-level vision–language tasks (Wang et al., 2024). As a general foundation model, GPT-5, exhibits strong cross-modal reasoning and, with instruction tuning and domain adaptation, can support medical VQA, report generation, and clinical decision support (Hou et al., 2025). Despite this progress, most prior work targets adult images, with limited coverage of obstetrics and fetal ultrasound, which is a basic tool for prenatal care. Multi-center heterogeneity and the complexity of multi-image/multi-view inputs remain open challenges. Overall, existing methods remain task-specific and confined to per-view analysis, whereas clinical practice requires aggregating information across multiple views to support diagnosis and decision-making. To the best of our knowledge, no existing AI model and dataset specifically address fetal ultrasound report generation and diagnosis.

**Fetal Ultrasound.** Ultrasound is the primary imaging modality for fetal anomaly screening, yet substantial appearance variability, scale differences, disease diversity, and multi-view images make automated interpretation challenging (Hu et al., 2023). Prior work has largely relied on supervised learning on single views, emphasizing standard-view recognition and automated biometry (Awadalla et al., 2023). In multi-image MLLM studies, Liu et al. employ DPO to guide models to attend to text-relevant regions across multiple images; however, these images often lack intrinsic inter-image dependencies. FetalCLIP learns anatomy-sensitive, generalizable representations via large-scale text–image contrastive learning and cross-modal alignment, benefiting downstream tasks such as classification and gestational-age estimation (Maani et al., 2025). The aforementioned works, e.g., FetalCLIP, operate at the level of single-image parsing within the clinical workflow (Figure 1c, *(1) Image-to-View*), focusing on view classification, organ segmentation, and a limited set of related sub-tasks, primarily to assist clinicians in identifying standard views. Beyond this, `FetalMind` is the first to achieve holistic interpretation of fetal ultrasound images and can directly generate full reports and diagnostic conclusions that support clinical decision-making.

## 3 CLINICAL FETAL ULTRASOUND DATASET CONSTRUCTION

In this section, we introduce the `FetalSigma-1M` dataset, composed of three subsets: ❶ Image–Report dataset: 20K image–report pairs, where each case includes multiple ultrasound images and a fine-grained clinical report covering biometric measurements, structural assessments, and abnormal findings. ❷ Image–Diagnosis dataset: 1M images organized as multi-image, case-level samples paired with physician-verified diagnostic reports. ❸ View Classification dataset: $10K$ fetal ultrasound images with fine-grained view annotations collected across three medical centers.

### 3.1 IMAGE–REPORT DATASET

**Scope & Scale.** We curate a large-scale, multi-center dataset for fetal ultrasound report generation and disease diagnosis that spans the full gestational spectrum and all fetal systems. The cohort comprises Early 5.0K, Mid 10.9K, and Late 5.2K examinations. Class balance is maintained with

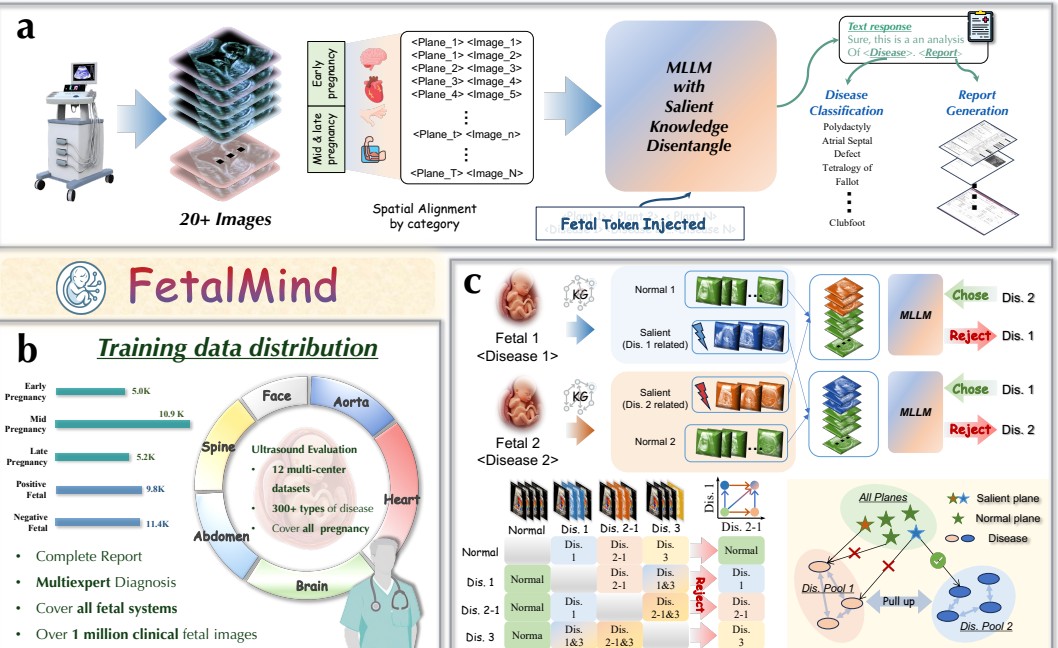

**Figure 3:** (a) `FetalMind` aligns with clinical cognition by classifying images into pregnancy-specific views, encoding disease–view keywords as special tokens, and reinforcing their intrinsic associations via salient epistemic disentanglement (SED). (b) `FetalSigma-1M` comprises 1 million fetal image–report–diagnosis triplets in 12 centers. (c) Overview of SED. Salient views are identified from disease–view graphs (see Section A.7) and treated as perturbation variables, swapped across fetuses with disease replaced. Bottom-left: intersection- and union-based substitution between diseased regions and views. Bottom-right: SVPO not only injects disease–view knowledge graphs into MLLMs but also enhances inter-disease discriminability

9.8K positive and 11.4K negative cases across 300+ disease categories. Data originate from 12 centers and multiple device models, totaling $> 1M$ clinical ultrasound images and enabling robust evaluation of cross-center generalization. Structured documentation across the heart, central nervous system, chest, abdomen, spine, face, neck, and long bones, covering all fetal systems, to support fine-grained fetal ultrasound analysis and multi-image modeling.

**Curation & Splits.** We apply unified multi-center cleaning, de-duplication, and quality control, including removal of low-quality frames and harmonization across devices/exports. Our survey (see Figure 1a) indicates that medical MLLMs trained on generic image–text pairs frequently miss diagnoses, which is an unacceptable failure mode in clinical practice. Accordingly, during curation we deliberately enriched positive cases to stabilize supervision, as routine fetal screening exhibits a base positive rate of $< 1\%$ in our observations across more than three centers. All positive case reports were finalized under the diagnoses of at least two expert clinicians.

## 3.2 IMAGE–DIAGNOSIS DATASET

Because many reports lack explicit diagnostic statements, we assigned a *Diagnosis* to each examination under physician supervision. Specifically, we constructed a disease ontology with 310 entities and their corresponding anatomical sites. Each report was then processed with DeepSeek-R1 (Guo et al., 2025) to extract provisional diagnoses by referencing this ontology, after which multiple expert fetal sonographers reviewed and corrected the outputs to obtain finalized diagnoses.

**View Classification dataset.** Accurate view localization from ultrasound video frames is the first step in fetal examination, as subsequent measurements and diagnoses rely on the correct anatomical view. Guidelines require nearly 20 standard views in the second trimester, with substantial variation across gestational stages and fetal positions, making automated modeling challenging (Salomon et al., 2022). To ensure reliable supervision, we annotated 11,358 images from three centers in `FetalSigma-1M` into 54 view categories under expert guidance, covering early, mid, and late gestation and including key views such as four-chamber, aortic arch, and three-vessel views. This subset is used to train the multi-view classification model in the Spatial Alignment stage (see Figure 1c).

# 4 METHODOLOGY

Figure 3a outlines how `FetalMind` is deployed within a fetal-ultrasound pipeline. Guided by clinical workflow, given multiple input images, `FetalMind` first performs *spatial alignment* to map each image to its anatomical view ($\triangleright$ Section 4.1), followed by *fetal token injection* to encode domain priors and mitigate disease confusion induced by text similarity ($\triangleright$ Section 4.2). We then describe how *view–disease swapping* constructs positive/negative pairs and how SVPO strengthens the model's preference for disease-relevant views ($\triangleright$ Section 4.3). Finally, we present the principles of multi-view swapping under different conditions. Please refer to Section B for more details.

## 4.1 CLASS-WISE SPATIAL ALIGNMENT

Identifying the correct imaging view is a prerequisite for reliable fetal diagnosis and report generation. To align with the view–image paradigm and remain robust against imaging noise, fetal pose variation, and gestational-age differences, we adopt a classification-based strategy. Given the substantial distribution shift between early, mid ,and late gestation, and the clinical practice of treating them as distinct tasks, we partition the 10K view-annotated images in `FetalSigma-1M` into *early* and *mid/late* subsets, using a 7:1:2 train/val/test split for pretraining. As illustrated in Figure 3a, the spatial alignment module incorporates two classifiers (Woo et al., 2023), trained separately on the double model. The early-gestation model spans 9 views categories, while the mid- and late-gestation model covers 41 categories, encompassing all clinically essential planes (Pellerito et al., 2018).

## 4.2 FETAL TOKEN INJECTION

We introduce the *Fetal Token Injection* strategy to explicitly encode domain-specific priors in fetal ultrasound by mapping key terms to spatial tokens. The rationale stems from the holistic nature of the fetus: although over 300 congenital anomalies have been documented, many exhibit highly similar linguistic descriptions (e.g., *ventricular septal defect* vs. *atrial septal defect*), yet correspond to clinically distinct diseases with divergent prognoses and management strategies. Similarly, prenatal ultrasound defines more than 40 standard imaging planes. While their textual descriptions may partially overlap, these planes are not interchangeable in clinical workflows. Without explicit token-level disentanglement, MLLMs tend to conflate semantically similar but clinically independent entities, ultimately yielding unreliable predictions and hallucinated report generation. This strategy introduces structured, view- and disease-aware tokens that enforce clear separability among near-synonymous terms and imaging planes, enhancing the reliability of diagnosis and reporting.

## 4.3 SALIENT EPISTEMIC DISENTANGLEMENT

Each fetus $i$ is represented as a multi-view sample $\mathcal{X}i = (p, I_{i,p}), p \in \mathcal{P}$, where $\mathcal{P}$ denotes the set of anatomical views and $Ii, p$ the image for view $p$. View–image correspondence $(p, I_{i,p})$ is obtained by the class-wise spatial alignment (Section 4.1). As shown in Figure 3c, the clinically confirmed disease set is $\mathcal{D}i \subseteq \mathcal{V}$dis. We construct an expert-curated disease→view *bipartite knowledge graph* $G : \mathcal{V}_{\text{dis}} \to 2^{\mathcal{P}}$ under the guidance of textbooks and experts that maps each disease $d$ to its salient views $G(d) \subseteq \mathcal{P}$. Given $d$, define the salient and non-salient view sets $\mathcal{P}^{(+)}(d) = G(d), \mathcal{P}^{(-)}(d) = \mathcal{P} \setminus G(d)$ and split $\mathcal{X}_i$ as $\mathcal{X}_i^{(+;d)} = \{(p, I_{i,p})\}_{p \in \mathcal{P}^{(+)}(d)}, \mathcal{X}_i^{(-;d)} = \{(p, I_{i,p})\}_{p \in \mathcal{P}^{(-)}(d)}$.

**View-Disease swap.** Pick two fetal cases $i \neq j$ with $d_i \in \mathcal{D}_i$, $d_j \in \mathcal{D}_j$, and $d_i \neq d_j$. We swap only the salient views aligned by the established view–image correspondence: $(p, I_{i,p})$:

$$\widetilde{\mathcal{X}}_{i \leftarrow j}^{(d_j)} \triangleq \mathcal{X}_i^{(-;d_j)} \cup \mathcal{X}_j^{(+;d_j)}\Big|_{\text{aligned by } (p, I_{i,p})}, \widetilde{\mathcal{X}}_{j \leftarrow i}^{(d_i)} \triangleq \mathcal{X}_j^{(-;d_i)} \cup \mathcal{X}_i^{(+;d_i)}\Big|_{\text{aligned by } (p, I_{j,p})}. \quad (1)$$

Let $x_i^{\text{swap}}$ and $x_j^{\text{swap}}$ denote the full inputs (images + prompt) built from equation 1, i.e., $x_i^{\text{swap}} \triangleq (\widetilde{\mathcal{X}}_{i \leftarrow j}^{(d_j)}, \text{prompt})$ and $x_j^{\text{swap}} \triangleq (\widetilde{\mathcal{X}}_{j \leftarrow i}^{(d_i)}, \text{prompt})$. Note that any change in the images during swapping requires a synchronized update of the prompt accordingly. Our goal is to *reject* the receiver's original disease set under swapped evidence. For each swapped input we form preference triplets: $(x_i^{\text{swap}}, \mathcal{D}_j, \mathcal{D}_i)$ and $(x_j^{\text{swap}}, \mathcal{D}_i, \mathcal{D}_j)$. The *chose* labels come from the donor and *reject* labels come from the receiver's labels. We collect all triplets into the swap-derived set $\mathcal{D}_{\text{swap}}$. Early and mid-to-late pregnancy stages are swapped independently to account for their morphological differences.

**Table 1:** Comparison of `FetalMind` with other MLLM and unified multi-modal models on medical visual comprehension tasks. **Bold** and underlined text indicates the best performance and second-best performance, respectively. Note that * indicates models fine-tuned with *Supervised Fine-Tuning* to ensure a fair comparison.

| Type | Model | #Params | Medical LVLM | NLG Metrics ↑ | | | | CE Metrics ↑ | | | ACC↑ | Body F1-20↑ | Avg. ↑ |
|------|-------|---------|--------------|------|------|------|------|------|------|------|------|-------------|--------|
| | | | | B-1 | B-4 | MTR | R-L | P | R | F1 | | | |
| **w/o US Train** | InternVL3 | 1B | ✗ | 13.5 | 2.6 | 2.3 | 7.4 | 0.0 | 0.0 | 0.0 | 46.2 | 0.0 | 8.9 |
| | QwenVL2.5 | 7B | ✗ | 7.8 | 1.4 | 1.2 | 3.9 | 13.0 | 0.5 | 1.0 | 46.8 | 2.5 | 8.7 |
| **w/ US Train** | BiomedGPT | 182M | ✓ | 1.6 | 0.3 | 0.7 | 1.2 | 3.5 | 1.6 | 1.9 | 46.8 | 5.9 | 6.9 |
| | LLaVA-Med | 7B | ✓ | 0.9 | 0.3 | 0.4 | 0.6 | 2.0 | 0.1 | 0.2 | 46.2 | 0.8 | 5.6 |
| | LLaVA-Med * | 7B | ✓ | 6.3 | 3.0 | 4.4 | 5.6 | 1.9 | 0.1 | 0.1 | 46.9 | 0.8 | 11.6 |
| | Med-Flamingo | 8.3B | ✓ | 21.6 | 8.9 | 8.5 | 7.7 | 3.8 | 1.1 | 1.7 | 44.1 | 1.6 | 14.5 |
| | Gemini 2.5 Pro | - | ✗ | 16.9 | 7.0 | 9.9 | 12.9 | 19.4 | 16.1 | 17.6 | 71.4 | 26.4 | 24.2 |
| | GPT-5 | - | ✗ | 28.3 | 8.3 | 4.8 | 12.4 | 19.1 | 12.6 | 15.2 | 71.6 | 23.6 | 24.1 |
| | InternVL3 * | 1B | ✓ | 14.1 | 4.0 | 4.9 | 6.5 | 26.2 | 18.9 | 22.0 | 78.2 | 39.9 | 23.9 |
| | `FetalMind`-S1 | 1B | ✓ | 30.3 | 9.2 | 15.5 | 12.4 | 23.1 | **29.2** | 25.8 | 79.0 | 45.2 | 29.7 |
| | `FetalMind`-M7 | 7B | ✓ | **33.9** | **23.1** | **30.4** | **30.7** | **34.7** | 28.2 | **31.1** | **81.3** | **50.2** | **38.2** |

**Data-Centric Learning via SVPO.** We optimize preference alignment on $\mathcal{D}_{\text{swap}}$ using Salient View Preference Optimization (SVPO). The key idea is a strategy that builds preference pairs by mining Salient Views from knowledge graph on top of existing preference-optimization algorithms. Either online rewards (e.g., PPO (Schulman et al., 2017)) or offline chosen/rejected pairs (e.g., DPO (Rafailov et al., 2024), CPO (Xu et al., 2024)) can be used; following prior visual alignment work (Yu et al., 2024a;b), we adopt the offline formulation. The SVPO objective is

$$\mathcal{L}_{\text{SVPO}}(\pi_\theta) = -\mathbb{E}_{(x,\mathcal{D}_w,\mathcal{D}_l)\sim\mathcal{X}}\left[\log\sigma\big(\beta\big(\log\pi_\theta(\mathcal{D}_w\mid x) - \log\pi_\theta(\mathcal{D}_l\mid x)\big)\big)\right], \quad (2)$$

where $\sigma$ is the sigmoid and $\beta>0$ is a temperature. Let the contrastive score be $g \triangleq \log\pi_\theta(\mathcal{D}_w\mid x) - \log\pi_\theta(\mathcal{D}_l\mid x), \Delta = \beta g$. The gradients are

$$\frac{\partial\mathcal{L}_{\text{prefer}}}{\partial\Delta} = \sigma(\Delta) - 1, \qquad \frac{\partial\mathcal{L}_{\text{prefer}}}{\partial g} = \beta\big(\sigma(\Delta) - 1\big). \quad (3)$$

When the chosen and rejected responses are very close ($\Delta\approx 0$, i.e., hard pairs), $\sigma(\Delta)\approx\frac{1}{2}$ and hence $\frac{\partial\mathcal{L}_{\text{prefer}}}{\partial g} \approx -\frac{\beta}{2}$, providing a non-negligible signal that *simultaneously* increases $\log\pi_\theta(\mathcal{D}_w\mid x)$ and decreases $\log\pi_\theta(\mathcal{D}_l\mid x)$. Consequently, SVPO naturally emphasizes hard pairs and sharpens fine-grained distinctions (*e.g.*, negation, units, laterality, anatomical loci) that are critical for medical report generation and diagnosis. As shown in Equation (2), SVPO reinforcement learning operates by constructing inputs $x$ and pairing them with chosen samples $\mathcal{D}w$ and rejected samples $\mathcal{D}l$. In our formulation, the training distribution is instantiated by the swap-derived dataset $\mathcal{D}_{\text{swap}}$.

**Principles of Swap Construction.** As shown in Figure 3c, we summarize four swap recipes for constructing preference pairs while preserving anatomical plausibility and inter-view consistency: ❶ **Disease-to-Normal.** Randomly sample two fetuses. For the receiver, remove disease-related images and replace them with the donor's *normal* images for the corresponding views. ❷ **Normal-to-Disease.** Sample a normal receiver and an abnormal donor. Replace the receiver's corresponding images with the donor's *disease-related* images; if a corresponding plane is missing, append the donor's disease-related plane set. ❸ **Disease-to-Disease.** Sample two abnormal fetuses with different disease. Remove the receiver's disease-related images and insert the donor's disease-related images to form a contrasted disease composition. ❹ **Disease Aggregation.** Sample two fetuses whose disease-related image sets are disjoint and merge them to synthesize a multi-disease case.

*Global constraints.* (1) Non-overlapping images are *kept from the receiver* rather than hallucinated. (2) When the number of images changes during a swap, the prompt must be updated accordingly.

## 5 EXPERIMENT

### 5.1 EXPERIMENTAL SETUP

**Benchmarks.** We randomly split data from nine centers into training/validation/test sets with a 7:1:2 ratio, and used data from the other three centers for external validation. To enable diverse

**Table 2:** Comparison of `FetalMind` with other LVLMs and unified multi-modal models on medical visual comprehension tasks. **Bold** and underlined indicates the best and second-best performance, respectively.

| Model | Early Preg. ↑ | | | | Mid Preg. ↑ | | | | Late Preg. ↑ | | | |
|---|---|---|---|---|---|---|---|---|---|---|---|---|
| | Micro-D | Macro-D | Micro-V | Macro-V | Micro-D | Macro-D | Micro-V | Macro-V | Micro-D | Macro-D | Micro-D | Macro-V |
| InternVL3 | 0.0 | 0.0 | 0.0 | 0.0 | 0.0 | 0.0 | 0.0 | 0.0 | 0.0 | 0.0 | 0.0 | 0.0 |
| QwenVL2.5 | 2.5 | 1.4 | 2.7 | 1.4 | 0.8 | 0.4 | 2.2 | 0.9 | 3.0 | 1.6 | 2.9 | 1.2 |
| BiomedGPT | 8.3 | 1.4 | 4.1 | 0.8 | 4.9 | 0.6 | 6.8 | 2.5 | 7.1 | 1.0 | 5.2 | 0.9 |
| LLaVA-Med | 0.9 | 0.2 | 0.8 | 0.2 | 0.5 | 0.1 | 1.2 | 0.2 | 0.3 | 0.1 | 0.0 | 0.0 |
| LLaVA-Med * | 0.4 | 0.1 | 0.4 | 0.1 | 0.1 | 0.0 | 1.1 | 0.2 | 0.7 | 0.1 | 0.5 | 0.1 |
| Med-Flamingo | 6.8 | 1.5 | 0.7 | 0.3 | 2.3 | 0.3 | 1.8 | 0.6 | 3.7 | 1.1 | 1.5 | 0.9 |
| Gemini 2.5 Pro | 20.5 | 13.8 | 21.4 | 19.6 | 19.5 | 16.2 | 27.2 | 17.2 | 24.5 | 17.7 | 27.4 | 16.5 |
| GPT-5 | 13.4 | 6.9 | 14.1 | 12.5 | 17.9 | 14.8 | 25.7 | 18.3 | 21.3 | 14.9 | 24.1 | 17.2 |
| InternVL3 * | 25.1 | 19.6 | 37.2 | 11.1 | 23.2 | 7.9 | 41.3 | 13.2 | 24.1 | 15.6 | 38.7 | 11.1 |
| `FetalMind-S1` | 25.8 | 30.7 | 27.8 | 18.5 | 30.2 | 19.3 | 47.9 | 21.6 | 36.9 | 30.2 | 44.5 | 18.1 |
| `FetalMind-M7` | **41.0** | **36.0** | **44.5** | **20.3** | **35.2** | **22.1** | **52.1** | **22.0** | **39.6** | **31.6** | **49.6** | **18.7** |

evaluation, we extract gestational-age metadata from ultrasound reports and partition the test set into *early*, *mid*, and *late* subsets, assessing robustness and generalization across stages. The evaluation results confirm the performance improvements of our model, particularly evident in early pregnancy diagnosis and major malformations. The metrics are provided in Section D.

**Baseline Methods.** We compare `FetalMind` against nine MLLM baselines. InternVL3 (Zhu et al., 2025) and QwenVL-2.5 (Bai et al., 2025) were not trained on ultrasound data. The other seven models incorporate ultrasound in their training pipelines, including BiomedGPT (Zhang et al., 2024), LLaVA-Med (Li et al., 2023), Med-Flamingo (Moor et al., 2023b), Gemini 2.5 Pro, GPT-5, and our SFT variants LLaVA-Med* and InternVL3* fine-tuned on `FetalSigma-1M`. For open-source models, we evaluate the released checkpoints using their official prompting strategies. Although Gemini 2.5 Pro and GPT-5 do not explicitly disclose prenatal ultrasound data, their stable performance and reported medical pretraining suggest indirect exposure; we therefore categorize them as *with-ultrasound* in our analysis. Note that for models lacking native diagnostic capability, we obtain the corresponding diagnoses by passing their generated reports to GPT (Guo et al., 2025), using carefully crafted prompts together with a structured specification of the disease set.

**Implementation Details.** We train the model on NVIDIA A800 GPUs with one epoch for the alignment stage, three epochs for instruction tuning, and one epoch for reinforcement learning with SVPO. The learning rate is set to $5 \times 10^{-5}$, and the temperature parameter is fixed at $\beta = 0.0$. Our 1B model is instantiated from InternVL3, whereas the 7B variant is built upon Qwen2.5-VL. For fairness, we fix the image size to $224 \times 224$ for all models. More results are provided in Section A.

## 5.2 EVALUATION ON GENERAL MULTI-CENTER STUDY

**Performance on Medical Diagnosis.** Medical diagnosis requires accurate prediction of one or more standardized labels, directly impacting clinical decision-making and patient outcomes. On the twelve-center disease-classification benchmark (Table 1), `FetalMind`-M7 improves binary abnormal/normal accuracy by $9.7\%$. Multi-label classification is particularly challenging for MLLMs because it demands disentangling subtle symptoms and mapping them to precise diagnoses. Under cross-entropy metrics, `FetalMind`-M7 achieves an F1 gain of $13.5\%$ and a recall gain of $9.3\%$ over prior models. To further assess localization fidelity from diseases to fetal anatomy, we construct a disease-view mapping spanning 20 anatomical categories (e.g., cardiac, cerebral). As shown in the penultimate column of Table 1, `FetalMind` achieves a $23.8\%$ gain, demonstrating the effectiveness of SED in grounding diseases to the correct images and reinforcing disease–view alignment.

**Performance on Medical Report Generation.** Medical report generation requires the model to generate a detailed report based on the provided medical scan. As shown in Table 1, `FetalMind`-M7 achieves the best scores, outperforming strong baselines (e.g., Gemini 2.5 Pro and GPT-5) by approximately **+5.6%** (BLEU-1), **+14.2%** (BLEU-4), **+20.5%** (METEOR), and **+17.8%** (ROUGE-L). The lighter `FetalMind`-S1 variant ranks second on most NLG metrics, indicating a favorable efficiency–performance trade-off. A visual comparison is provided in Figure 4. These gains suggest that SVPO encourages explicit correspondences between multiple images and diagnostic labels rather than treating images and labels as an undifferentiated set (see Figure 1b), thereby improving multi-image grounding and robustness for report generation and multi-label disease classification.

**Performance on Different Stages of Pregnancy.** Mastery of fetal ultrasound by physicians typically requires **3+ years** of education, considerably longer than X-ray interpretation (about 1 year),

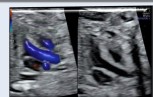

**Case #12388**

**User Query:** 1. <image> 2. <image> 3. <image> 4. <image> 5. <image> 6. <image> 7. <image> 8. <image> 9. <image> 10. <image> 11. <image> 12. <image> 13. <image> 14. <image> 15. <image> 16. <image> 17. <image> 18. <image> Please review each image in sequence (anatomical site annotated) and, based on all the images, summarize the FINDINGS and IMPRESSION.

**GT: Ventricular septal defect**

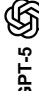

**GPT-5**

1) Single live fetus, cephalic presentation. Based on BPD/HC/AC/FL/HL, estimated gestational age ~33–34 weeks, consistent with late pregnancy; no definite sonographic evidence of growth restriction or major malformation. Amniotic fluid subjectively normal, placenta anterior wall with no abnormal appearance. 2) Fetal cardiac structure and hemodynamic ultrasound findings essentially normal: heart on the left, cardiac axis ~36°; four-chamber view symmetrical, atrioventricular valve offset present; LVOT, RVOT, and great vessel crossover visualized, aortic and pulmonary artery diameters proportionate; Color Doppler shows no significant valvular regurgitation or cross-septal shunt; fetal heart rate and rhythm normal; three-vessel trachea plane without abnormal arrangement or disproportion.

**Diagnosis : Normal**

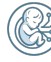

**FetalMind**

FINDINGS: Ultrasound description:[Obstetric Doppler Ultrasound] Number of fetuses: 1; Fetal presentation: cephalic; Fetal heart and movements: visible. Biometric measurements: BPD 88 mm, HC 309 mm, AC 291 mm; FL 63 mm, HL 55 mm. Placenta location: posterior wall; Placental thickness: 32 mm; Placental maturity: grade II. Amniotic fluid index: 29–25–33–28 mm. × 14.6 mm; Right atrium: 15.9 × 14.3 mm. Aorta: 7.2 mm; Pulmonary artery: 7.3 mm. Great vessel crossover present. [Cardiac M-mode Ultrasou [Fetal Echocardiography] Cardiac position: left chest; Apex orientation: left anterior. Abdominal circumference plane: descending aorta on the left, stomach bubble on the left. Cardiothoracic ratio = 0.52, cardiac axis 35.9°. [2D Echocardiography] Left ventricle: 23.9 × 13.7 mm; Right ventricle: 21.3 × 14.9 mm; Left atrium: 18.1 nd] Heart rate: 138 bpm, regular. Atrial contraction and relaxation regular; Ventricular contraction and relaxation regular. Atrioventricular rhythm consistent.
IMPRESSION: Possible ventricular septal defect; increased aortic flow velocity.

**Diagnosis : Ventricular septal defect**

**Figure 4:** Illustration of `FetalMind` versus GPT-5 on a representative case (ID: 12388). The ground-truth diagnosis is a ventricular septal defect (VSD). GPT-5 misclassified the case as normal, likely due to its limited utilization of 2D and Doppler signals. In contrast, `FetalMind` correctly identified the VSD by integrating multi-view structural cues with blood-flow features. The report is truncated for brevity.

underscoring the task's complexity. Following clinical practice, we stratify evaluation by gestational stage (*early, mid, late*) and report performance per trimester. As shown in Table 2, **Micro-D** denotes multi-label disease classification, while **Micro-V** measures performance after mapping diseases to anatomical regions. `FetalMind`-M7 surpasses all baselines across trimesters, with gains ranging from **2.2%** to **24.9%**, demonstrating strong generalization. Notably, in the *early* trimester, Micro-D improves by **20.5%**, highlighting the model's value for earlier detection of fetal anomalies—enabling earlier, potentially actionable findings and affording more time for follow-up and clinical decision-making. More experiments in Section A.6.

## 5.3 EVALUATION ON THE NINE MAJOR MALFORMATIONS

To assess the model's diagnostic capability for critical conditions, we curated 153 clinically confirmed cases covering nine major congenital anomalies, which are critical in prenatal ultrasound diagnosis in China, where misdiagnosis often leads to severe medical or legal consequences. These challenging cases were collected across three centers and multi-device models, providing clinically reliable ground-truth labels for evaluation. As shown in Figure 5, , GPT-5 and Gemini 2.5 Pro, despite being state-of-the-art MLLMs for fetal ultrasound, consistently failed to identify these anomalies and often misclassified them as negative. In contrast, `FetalMind` achieved a diagnostic accuracy of **98%**, substantially surpassing all prior baselines across anomaly types and demonstrating robust decision support in complex clinical settings.

**Figure 5:** Diagnostic performance comparison in nine major malformations

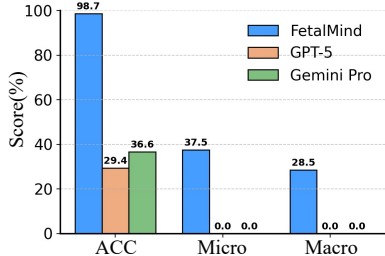

## 5.4 ABLATION STUDIES

**Ablation Studies on Strategy.** As shown in Table 3, removing any of the three components with *token injection*, *spatial alignment*, and *SVPO* degrades performance. We summarize three key observations: **Obs.❶** Eliminating fetal token injection yields the smallest yet consistent drop across all metrics. This indicates that injecting fetal priors at the token level mainly strengthens fine-grained discrimination and stability, enabling the model to separate semantically similar but clinically distinct entities. **Obs.❷** Removing spatial

**Table 3:** Ablation study on `FetalMind` in the `FetalSigma-1M` dataset. The impact of without (w/o) and with (w) post-selection techniques.

| Setting | B-4 | F1 | ACC | AVG |
|---|---|---|---|---|
| `FetalMind` | **23.1** | **31.1** | **81.3** | **45.2** |
| w/o Token inject | 21.9 | 30.7 | 80.3 | 44.3 |
| w/o Spatial align | 16.3 | 29.4 | 80.6 | 42.1 |
| w/o SED | 13.7 | 26.7 | 80.1 | 40.5 |
| w/ GRPO | 9.7 | 24.2 | 79.2 | 37.3 |
| w/ DPO | 7.9 | 12.3 | 65.8 | 28.7 |
| Vanilla | 9.2 | 25.8 | 79.0 | 38.0 |

alignment disproportionately reduces report generation quality while having a milder impact on diagnostic metrics. This suggests that cross-view spatial alignment primarily facilitates multi-image aggregation and narrative coherence, effectively multiple views into a *clinically interpretable* sum-

mary. **Obs.❸** Removing SED causes the largest overall decline, establishing it as the primary source of improvement. By aligning multi-view preferences, SED simultaneously enhances report readability and stabilizes diagnostic discrimination, underscoring its central role in multi-view reasoning.

**Ablation Studies on Reinforcement Learning.** We further investigate the effect of different *reinforcement learning objectives* in Table 3. Compared with vanilla training, , models optimized with DPO (Rafailov et al., 2023) or GRPO (Shao et al., 2024) perform worse across BLEU-4, F1, and ACC. In contrast, `FetalMind` achieves the strongest overall results. These findings underscore the importance of the post-selection procedure and demonstrate that SVPO with salient epistemic disentanglement is essential for enhancing diagnostic accuracy and producing clinically faithful reports.

### 5.5 PARAMETER SENSITIVITY ANALYSIS

**Temperature** $\beta$. As shown in Figure 6, we observe a distinct task-dependent trend. For *diagnostic classification*, lower temperatures consistently yield stronger performance, as reduced sampling stochasticity improves label consistency and raises F1/ACC. In contrast, for *report generation*, a mild degree of randomness proves beneficial: performance peaks around $\beta = 0.1$, balancing exploratory diversity with factual stability. These results suggest a near-deterministic setting for diagnosis and a small but nonzero temperature for narrative generation.
**Report Generation vs. Diagnosis.** `FetalMind` highlights an inherent heterogeneity between report generation and diagnostic classification in both task objectives

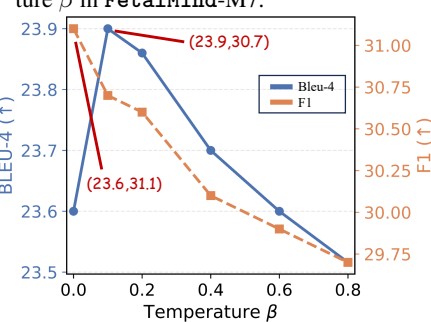

**Figure 6:** Parameter sensitivity of temperature $\beta$ in `FetalMind`-M7.

and evaluation metrics. As shown in Figure 5, excessive determinism and insufficient randomness reduce report coverage and completeness. Enabling *controlled exploration* in lesion-related segments while preserving determinism for diagnostic-critical points, and adopting task-specific, temperature-aware inference, further improves overall performance.

## 6 DISCUSSION

`FetalMind` achieves best performance on both fetal report generation and diagnostic, surpassing both general large models and domain-specific medical models. An insight emerges: structured tool usage in medical AI holds value. Compared with purely end-to-end methods, coupling the reasoning capacity of large models with domain basic modules consistently yields superior performance.

**Generalists Versus Specialists.** A notable finding is that general-purpose models (e.g., GPT-5, Gemini 2.5 Pro) overall outperform specialized medical models (e.g., LLaVA-Med (Li et al., 2024), Med-Flamingo (Moor et al., 2023b)). This indicates that narrow specialization may diminish the broad reasoning abilities conferred by large-scale pretraining. By integrating domain-specific tools under clinical guidance, `FetalMind` provides an effective bridge between the two paradigms.

**Limitations & Future Work.** Our evaluation remains retrospective and constrained by the available dataset, and prospective clinical studies are crucial for establishing real-world utility and safety. On the other hand, there remains a theoretical risk that the model may inadvertently learn "splicing artifacts" from synthetic data. Promising directions include: (1) tighter integration with PACS and ultrasound consoles for seamless clinical deployment; (2) uncertainty estimation and case triage to enhance clinician oversight; (3) broader coverage of rare anomalies and robustness to domain shift through active and continual learning; (4) privacy-preserving federated training across hospitals; and (5) extending disease–view graphs to temporal modalities. We anticipate that `FetalSigma-1M` and `FetalMind` will catalyze clinically grounded research toward trustworthy fetal ultrasound AI.

## 7 CONCLUSION

In this work, we present `FetalMind`, a clinically guided AI system for fetal ultrasound and, to our knowledge, the first unified framework addressing both report generation and diagnosis. By incorporating bipartite graph and disentangling disease–view heterogeneity, our SED aligns the model's reasoning trajectory with real-world diagnostic workflows. Trained on the newly curated `FetalSigma-1M` comprising 20K reports from 12 centers, `FetalMind` consistently outperforms both open-source and proprietary baselines across all gestational stages. Beyond improvements, our findings underscore the critical role of structured clinical priors in building reliable AI systems.

## REPRODUCIBILITY STATEMENT

To ensure the reproducibility of this research, we describe the experimental setup, data processing steps, and key implementation details. Specifically, we employed reinforcement learning from the MS-Swift framework and used LLaMA-Factory for supervised fine-tuning, with all implementations developed in PyTorch. The datasets used in this work are derived from real clinical applications; a subset of the reports is included in the paper, and we will also release the trained model weights.

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

APPENDIX

In this appendix, we provide supplementary material to further elucidate our approach. Section A expands on the experiments with detailed protocols and ablation studies. Section B introduces the preliminaries of the Salient Epistemic Disentanglement (SED) reinforcement learning module. Section C visualizes the standardized structured report template that guides fetal ultrasound report generation and diagnosis. Finally, Section D consolidates the evaluation metrics and their definitions used throughout the paper.

# A    MORE EXPERIMENTS

## A.1    ATTENTION ANALYSIS

**Implementation Details.** We curate a total of 10,000 SVPO samples, with approximately 2,500 assigned to each of the four states. To mitigate confounding due to inter-institution variability, SED construction is restricted to within-center data. This choice is motivated by two practical considerations: ❶ report templates vary substantially across medical centers, introducing formatting and phrasing biases; and ❷ for a given fetus, all images are acquired on the same device at the same site. Constraining SED to a single center therefore attenuates center/device effects and yields a cleaner evaluation of SVPO behavior. `FetalMind` was trained using data from multiple devices, including 15 types of ultrasound machines from over four manufacturers.

To evaluate whether our proposed SED module indeed guides the model to focus on pathological regions after training, we conducted a quantitative attention analysis. Following the design in Figure 2 *Left*, we computed the MeanALLQ, defined as the mean attention weight over all query tokens across layers and heads, for both abnormal and normal ultrasound images. We then examined how often the attention allocated to abnormal images dominates that of normal images, thereby reflecting the model's capacity to capture clinically salient cues. As summarized in Table 4, the baseline Qwen2.5-VL model achieves a dominance ratio of only 39.1% (713/1824). Incorporating additional training signals (Qwen2.5-VL*) improves this ratio to 52.4% (956/1824). In contrast, our `FetalMind`-M7 substantially outperforms both baselines, with abnormal images receiving higher attention weights in 80.7% of cases (1472/1824). These results clearly indicate that SED effectively enhances the model's ability to attend to pathological regions, thus strengthening its diagnostic reliability.

**Table 4:** Ratio-based evaluation of attention dominance on salient images. The *Salient* denotes the number of abnormal cases with higher MeanALLQ values than normal cases, while the *Normal* is the total number of test cases. Percentages reflect the proportion of salient images receiving stronger attention. (*) indicates models further tuned with supervised fine-tuning (SFT).

| Model | Salient | Normal | Percentage |
|---|---|---|---|
| Qwen2.5-VL-7B | 713 | 1824 | 39.1% |
| Qwen2.5-VL-7B * | 956 | 1824 | 52.4% |
| `FetalMind`-M7 | 1472 | 1824 | 80.7% |

## A.2    CONFUSION MATRIX

To further investigate the robustness of our framework and the fidelity of generated reports, we conducted additional retrospective evaluations involving clinical experts. Specifically, we compared two strong vision–language baselines, **Gemini 2.5 Pro** and **GPT-5**, alongside our method, to examine whether evaluators could distinguish model-generated reports from authentic clinical reports.

Figure 7 presents the aggregated confusion matrix across all 12 medical centers. Notably, evaluators often misclassified reports generated by large models as authentic, indicating that both Gemini 2.5 Pro and GPT-5 achieved a high level of realism in language style and clinical adequacy. Nevertheless, GPT-5 exhibited slightly higher indistinguishability, suggesting stronger alignment with clinical reporting conventions.

To further assess robustness under physiological heterogeneity, we stratified the evaluation by gestational stages. As illustrated in Figure 8, evaluator performance remained consistent across early-,

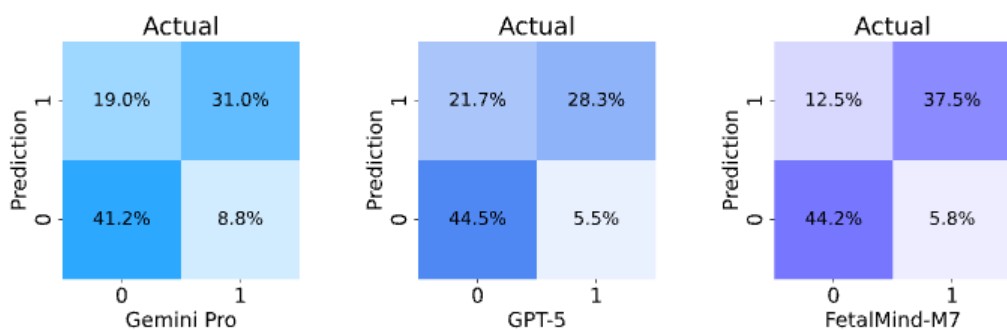

**Figure 7:** Confusion matrix for evaluators to identify reports generated by large models in the retrospective study, covering results from all 12 medical centers.

mid-, and late-gestation groups. The relative advantage of GPT-5 over Gemini 2.5 Pro persisted across all stages, reinforcing the conclusion that larger-scale alignment contributes to improved cross-condition fidelity. These findings collectively support the reliability of our framework and highlight the competitive performance of cutting-edge foundation models when benchmarked under rigorous human evaluation.

### A.3 REPORT GENERATION STUDY

To further substantiate the effectiveness of our approach, we include a representative case study in Figure 9. In this example (Case #127858), the ground-truth diagnosis is *skeletal dysplasia*. While GPT-5 misclassifies the case as normal, `FetalMind` correctly identifies the pathology by jointly exploiting multi-view anatomical context and Doppler flow cues. This case illustrates how injecting domain-specific priors and explicitly modeling cross-view correspondences enables the system to recover subtle abnormalities that general-purpose LVLMs often overlook, thereby improving diagnostic reliability in fetal ultrasound.

### A.4 GESTATIONAL AGE DISTRIBUTION

In addition to evaluator-based assessments, we also analyzed the distribution of gestational ages across centers in `FetalSigma-1M`. This is important because fetal ultrasound exhibits substantial heterogeneity in image appearance and reporting style at different stages of pregnancy, which may confound both training and evaluation if not carefully accounted for. Figure 10 shows the gestational age distributions extracted from three representative medical centers. Clear differences in case composition can be observed: while one center contributes a larger proportion of early-gestation cases, others are skewed toward mid-to-late gestation. Such heterogeneity motivates our stage-wise stratification strategy and provides empirical justification for evaluating model robustness under diverse physiological regimes. These analyses further highlight the challenges of building foundation models for fetal ultrasound and underline the necessity of multi-center, stage-aware evaluation.

### A.5 REPORT CLASSIFICATION

**Fetal Ultrasound Report Classification** To validate the effectiveness of `FetalMind`, we introduce an ablation experiment where the model classifies fetal ultrasound reports based on a list of predefined disease labels. The process begins with the model generating a report from the ultrasound data, followed by selecting relevant disease labels based on the report's content. The selected labels are then compared to the ground truth labels provided by clinical experts. The final classification accuracy is used to assess the model's performance across several benchmarks. Our findings indicate that `FetalMind` offers a significant improvement in both diagnostic accuracy and clinical relevance compared to previous approaches. The prompt used to guide the model in classifying the ultrasound report is as follows:

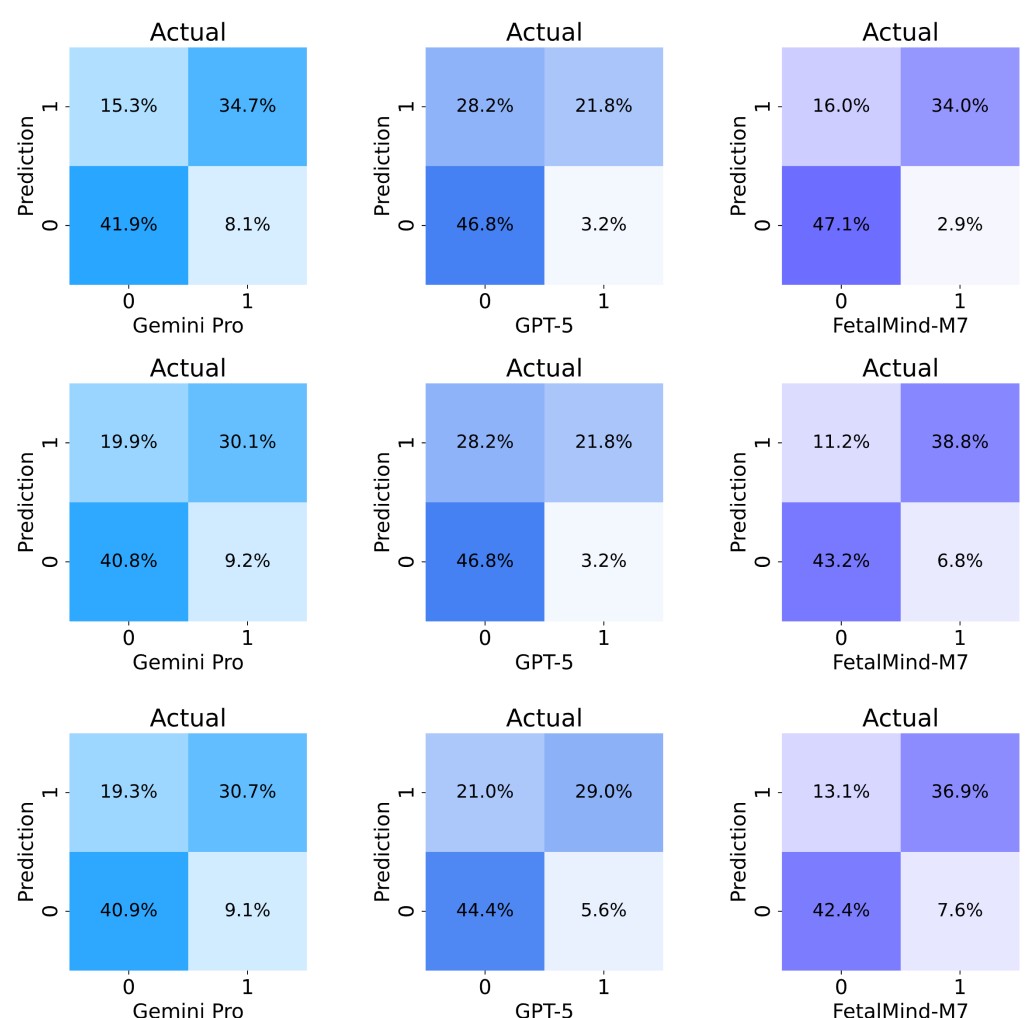

**Figure 8:** Confusion matrices illustrating evaluator performance in distinguishing reports generated by large language models during the retrospective study. Results are stratified by early-, mid-, and late-gestation stages, reflecting variability across different phases of pregnancy and highlighting the consistency of evaluation outcomes under diverse clinical conditions.

You are an expert in fetal ultrasound diagnosis. Based on the following ultrasound report, please select the disease labels that are explicitly mentioned or can be definitively inferred. The disease labels are provided in a predefined list.

The specific requirements are as follows:

1. Only select labels that are directly related to the content of the report.

2. If there are multiple disease labels, separate them with commas.

3. The output should be formatted as: Disease1, Disease2, ... (do not include numbering, explanations, or quotation marks).

4. If no disease labels are relevant, return an empty string.

Please review the report and select the disease labels accordingly.

**Available Disease Labels:** {Label1, Label2, Label3, ...}

**Case #127858**

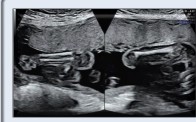

**User Query:** 1. <image> 2. <image> 3. <image> 4. <image> 5. <image> 6. <image> 7. <image> 8. <image> 9. <image> 10. <image> 11. <image> 12. <image> 13. <image> 14. <image> 15. <image> 16. <image> 17. <image> 18. <image> Please review each image in sequence (anatomical site annotated) and, based on all the images, summarize the FINDINGS and IMPRESSION.

**GT: Skeletal dysplasia**

**FetalMind**

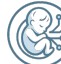

FINDINGS: Ultrasound description:
Number of fetuses: 1; Fetal presentation: cephalic; Fetal heart activity and movements: visible.
Biometric measurements: BPD 94 mm, HC 330 mm, AC 330 mm; FL 62 mm, HL 56 mm.
Placenta location: anterior wall; Placental thickness: 32 mm; Placental maturity: grade II+.
Lower placental margin >2 cm from the internal cervical os.
Amniotic fluid index: 51–35–23–23 mm.
Color Doppler: intracardiac blood flow detected.
Umbilical artery: PI 0.95, RI 0.61, S/D 2.57.
FHR: 143 bpm.
U-shaped notch observed at the neck.

**Diagnosis : Skeletal dysplasia**

IMPRESSION: Singleton pregnancy, cephalic presentation. Fetal femur length < –2 SD..

**GPT-5**

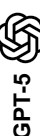

FINDINGS:
•Examination method: Transabdominal fetal ultrasound (partial planes/parameters).
•Number of fetuses: Singleton (measurements and anatomy shown correspond to a single fetus).
•Cranial findings: Transverse view shows regular skull contour, midline structures identifiable, no obvious cranial shape abnormalities. Biparietal diameter (BPD) approximately 8.3 cm (onscreen measurement), corresponding to gestational age ~32–33 weeks.
•Long bones of the limbs:
  • Femur length (FL): 61.8 mm (onscreen measurement clearly displayed).
  • Other long bones shown (limb long bone planes) measure approximately 58–66 mm; cortical continuity preserved, bone echogenicity and morphology without obvious bowing, shortening, or angulation deformities; no signs of fracture.
•Other systems/parameters: Current images do not display placental position or maturity, amniotic fluid volume, cardiac structures or rhythm, abdominal organs, umbilical cord and Doppler parameters, or cervix; therefore, these cannot be evaluated.
Overall biometric impression:
•Based on BPD ≈ 8.3 cm and FL = 61.8 mm, estimated gestational age is approximately 32–33 weeks, with general concordance between the two.
•Due to absence of complete head circumference (HC), abdominal circumference (AC), and integrated formulas, estimated fetal weight (EFW) is not calculated.

**Diagnosis : Normal**

IMPRESSION:
1.Singleton intrauterine pregnancy. No obvious abnormalities of fetal skull or long bones in the planes provided; measured BPD ≈ 8.3 cm and FL = 61.8 mm, overall growth consistent with gestational age of ~32–33 weeks.
2.As only partial images and parameters are available, placental characteristics, amniotic fluid, cardiac and great vessel structures, abdominal organs, and umbilical cord/Doppler cannot be assessed. Recommend correlation with a complete systematic screening or growth monitoring report; if clinically indicated, perform comprehensive fetal echocardiography and Doppler assessment, and follow up with serial growth parameters.

**Figure 9:** Illustration of `FetalMind` and GPT-5 Case Study. (Case 127858) Correct answer is skeletal dysplasia. GPT-5 misclassified it as normal, while `FetalMind` correctly identified skeletal dysplasia by integrating multi-view structures and blood flow features.

**Table 5:** Overall comparison of NLP and classification metrics between Doctor and Doctor+AI.

| Method | BLEU-1 | BLEU-4 | ROUGE-1 | ROUGE-L | METEOR | Precision$_{\text{micro}}$ | F1$_{\text{micro}}$ |
|---|---|---|---|---|---|---|---|
| Doctor | 75.388 | 67.817 | 77.450 | 72.019 | 40.592 | 0.568 | 0.562 |
| Doctor+AI | **88.532** | **81.605** | **86.002** | **85.717** | **59.351** | **0.679** | **0.653** |

**Ultrasound Report:** {[Insert ultrasound report here]}

Please provide the disease labels in the format mentioned above.

## A.6  REAL-WORLD CLINICAL DECISION-MAKING ANALYSIS

To further validate the effectiveness of our method, we conducted a real-world clinical scenarios test on 56 cases from two centers (as shown in Table 5). Specifically, we set up three groups: one with a

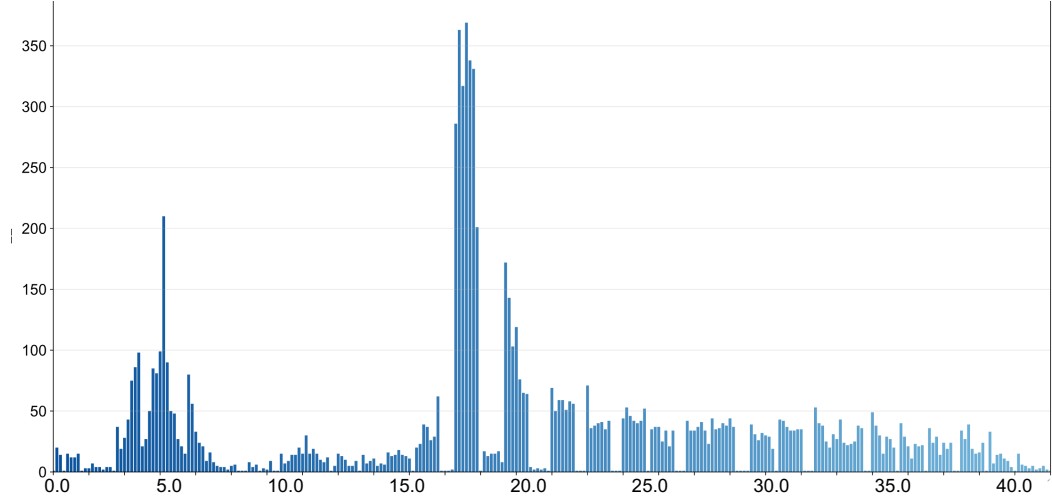

**Figure 10:** Visualization of gestational age distributions extracted from three medical centers. The figure highlights differences in case composition across centers, providing insights into data heterogeneity and supporting stratified analyses in subsequent model training and evaluation.

moderately experienced doctor, one with a moderately experienced doctor + AI, and a real clinical control group (three doctors including at least one highly experienced doctor). After completing the image collection for the examination, diagnosis was performed simultaneously, as shown in Table 1. As can be seen, our `FetalMind` can assist doctors by improving the neatness of the report writing and enhancing diagnostic accuracy.

### A.7 VISUALIZATION OF THE DISEASE–VIEW GRAPH

In Figure 11, we present a visualization of the Disease–View graph using a Sankey diagram. This method effectively represents the relationships between different diseases and body regions, with the flow width indicating the intensity or frequency of each connection. Our disease–view bipartite graph contains 326 disease nodes, 54 view nodes, and 879 corresponding edges. All nodes are determined based on textbooks, clinical guidelines, and expert consensus, and subsequently standardized through unified terminology to ensure consistency. We further detail the expert-in-the-loop construction process: three clinicians with over 10 years of experience reviewed the preliminary disease–view relations, refined them, and conducted multiple rounds of discussion. For cases where expert opinions diverged, we clarify in the revised manuscript that the resolution followed a Delphi-style anonymous voting procedure or arbitration by a senior third expert.

## B PRELIMINARY AND ANALYSIS

### B.1 PRELIMINARY

To improve an LVLM's reasoning over *multi-image* inputs, we adopt *visual preference alignment*. This section formalizes the objective and uses *CPO* as a representative instantiation.

**Visual Preference Alignment** Preference alignment trains a model so that its output preferences conform to human (or proxy) preferences. Prominent paradigms include **R**einforcement **L**earning from **H**uman **F**eedback (**RLHF**) (Ouyang et al., 2022) and **R**einforcement **L**earning from **AI** **F**eedback (**RLAIF**) (Bai et al., 2022). Let a dataset $D$ consist of triplets $\{x, y_w, y_l\}$,[1] where $x$ is a multimodal prompt—an interleaved sequence of images $v$ and texts $t$—and $y_w/y_l$ denote the *chosen* and *rejected* responses, respectively. Given a policy $\pi_\theta(y \mid x)$ and a reward model $r(x, y)$ that assigns higher scores to preferred responses, the visual preference alignment objective maximizes

---

[1] For clarity we present single-sample notation; the extension to mini-batches is straightforward.

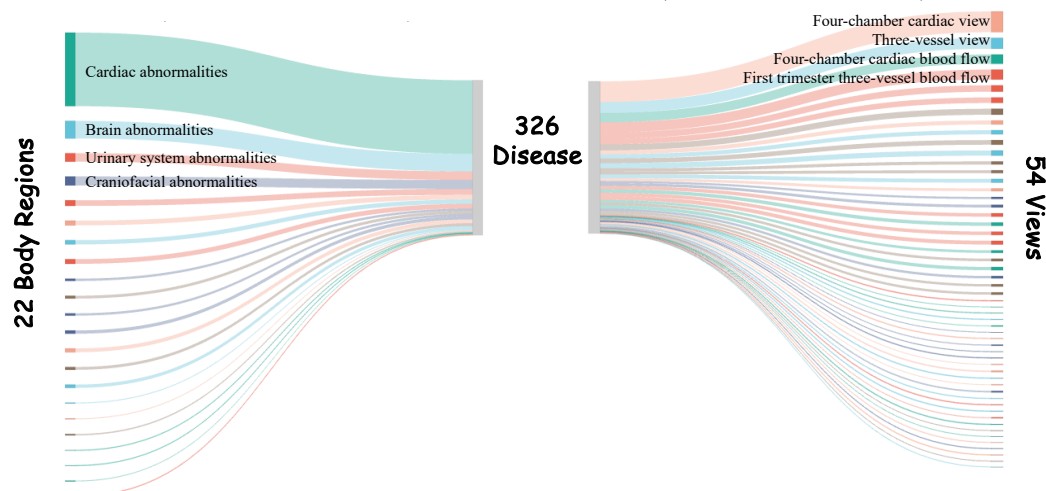

**Figure 11:** Visualization of Disease–View bipartite graph using a Sankey diagram. **Body Regions** represent different parts of the fetus, including the head, heart, and others.

expected reward:

$$\max_{\theta} \; \mathbb{E}_{x \sim D, \, y \sim \pi_\theta(y|x)} [r(x, y)], \tag{4}$$

where $\theta$ parameterizes the LVLM. To mitigate overfitting and constrain drift from a reference policy $\pi_{\mathrm{ref}}$, one augments the objective with a KL regularizer:

$$\max_{\theta} \; \Big[ \mathbb{E}_{x \sim D, \, y \sim \pi_\theta(y|x)} [r(x, y)] - \beta \, D_{\mathrm{KL}} \big( \pi_\theta(y \mid x) \, \| \, \pi_{\mathrm{ref}}(y \mid x) \big) \Big], \tag{5}$$

where $\beta > 0$ balances reward maximization and policy proximity. In practice, $\pi_{\mathrm{ref}}$ is the model snapshot before preference alignment.

**CPO contrastive score** CPO instantiates preference learning via a contrastive margin between the chosen and rejected responses:

$$\Delta \; = \; \beta(\log \pi_\theta(y_w \mid x) \, - \, \log \pi_\theta(y_l \mid x)), \qquad \mathcal{L}_{\mathrm{prefer}} \; = \; -\log \sigma(\Delta), \tag{6}$$

where $\sigma(\cdot)$ is the logistic sigmoid and $\beta > 0$ acts as a temperature.

**Near-tie behavior (hard pairs)** Let $g \triangleq \log \pi_\theta(y_w \mid x) - \log \pi_\theta(y_l \mid x)$ so that $\Delta = \beta g$. The gradients are

$$\frac{\partial \mathcal{L}_{\mathrm{prefer}}}{\partial \Delta} \; = \; \sigma(\Delta) - 1, \qquad \frac{\partial \mathcal{L}_{\mathrm{prefer}}}{\partial g} \; = \; \beta \big( \sigma(\Delta) - 1 \big). \tag{7}$$

For *hard* pairs where the two responses are nearly tied ($\Delta \approx 0$), we have $\sigma(\Delta) \approx \frac{1}{2}$ and thus

$$\frac{\partial \mathcal{L}_{\mathrm{prefer}}}{\partial g} \; \approx \; -\frac{\beta}{2}, \tag{8}$$

yielding a substantial, stable signal that simultaneously increases $\log \pi_\theta(y_w \mid x)$ and decreases $\log \pi_\theta(y_l \mid x)$. This property encourages fine-grained discrimination among near-synonymous or subtly different responses—e.g., negation, units, laterality, or anatomical loci—crucial for medical report generation and diagnosis from multi-view ultrasound.

**Difference from DPO** DPO also optimizes a margin, but it uses a *reference-adjusted* form

$$\tilde{\Delta} \; = \; \beta \left[ \log \frac{\pi_\theta(y_w \mid x)}{\pi_{\mathrm{ref}}(y_w \mid x)} - \log \frac{\pi_\theta(y_l \mid x)}{\pi_{\mathrm{ref}}(y_l \mid x)} \right], \tag{9}$$

which entangles the learning signal with the quality and stylistic biases of $\pi_{\mathrm{ref}}$ and typically incurs additional compute/memory overhead. In contrast, CPO's margin depends solely on $\pi_\theta$, delivering a cleaner, reference-free signal on near-ties and promoting a more compact, clinically faithful chosen-response distribution for multi-image inputs.

**Table 6:** Ablation study on `FetalMind` in the `FetalSigma-1M` dataset. The impact of without (w/o) and with (w) post-selection techniques.

| Setting | B-4 | F1 | ACC | AVG |
|---|---|---|---|---|
| FetalMind | **23.1** | **31.1** | **81.3** | **45.2** |
| w/o SED | 13.7 | 26.7 | 80.1 | 40.5 |
| w/ GRPO | 9.7 | 24.2 | 79.2 | 37.3 |
| w/ DPO | 7.9 | 12.3 | 65.8 | 28.7 |
| Vanilla | 9.2 | 25.8 | 79.0 | 38.0 |

## B.2 ANALYSIS OF SVPO AND SED

**SVPO** is readily extensible to multimodal large language models (MLLMs) that perform multi-image reasoning, and can be seamlessly integrated into multi-image inference pipelines according to task requirements. By explicitly distinguishing salient from non-salient images, SVPO improves both computational efficiency and predictive accuracy. For example, in a three-image joint analysis scenario where the key evidence primarily resides in Image 1, SVPO effectively steers the model toward the most informative visual cues.

**SED** introduces graph-aware reasoning by leveraging a bipartite graph to separate salient from non-salient images. When combined with SVPO, SED further establishes preference relations between abnormal images and target conditions, allowing condition–image structural information to be naturally injected into the MLLM's reasoning process. This design closely mirrors clinical workflows, where physicians select key views, focus on abnormal regions, and integrate disease knowledge across multiple images. Consequently, the framework is particularly well-suited to multi-image reasoning tasks with explicit graph-structured relationships.

In summary, SED embeds and strengthens SVPO, enabling the model not only to capture saliency relationships across images, but also to perform condition–image relational reasoning via graph structures. This yields a more principled foundation for interpretability and reliability in structured medical applications of MLLMs.

## B.3 ANALYSIS OF REINFORCEMENT LEARNING METHODS

**DPO performance drop causes and the role of CPO's BC regularizer.** DPO relies on a fixed reference policy $\pi_{\mathrm{ref}}$ (typically an SFT model) and optimizes a preference loss of the form $L(\pi_\theta; \pi_{\mathrm{ref}})$. This implicitly constrains $\pi_\theta$ to stay close to the reference, which can be suboptimal in our setting: in specialized domains such as medicine, the reference model often underfits domain-specific knowledge, and hard-anchoring $\pi_\theta$ to such a reference can limit the achievable performance.

In contrast, CPO removes this dependence by setting $\pi_{\mathrm{ref}} = U$, i.e., a uniform reference, and directly optimizing the contrastive objective $L(\pi_\theta, U)$. This design allows the policy to move beyond the limitations of the reference model and better align with the preference data. However, a purely contrastive preference loss $L(\pi_\theta, U)$ only encodes *relative* signals ("$y_w$ is preferred over $y_l$"), without constraining the *absolute* likelihood of preferred outputs. As a result, optimizing only the preference term can drive the model toward over-emphasizing superficial or stylistic characteristics of the preferred responses, rather than preserving factual correctness and faithfulness. In other words, the model may learn that "more elaborate / more confident / more verbose" is preferred, and lean toward "stylistic enhancement" instead of robustly modeling the underlying target distribution.

To address this, we introduce a BC regularizer

$$\mathbb{E}_{(x,y_w)\sim D} KL\big(\pi_w(y_w|x) \,\|\, \pi_\theta(y_w|x)\big) < \epsilon,$$

which, as shown in Appendix C, is equivalent to adding a negative log-likelihood term on the preferred data:

$$L_{\mathrm{CPO}}(\pi_\theta) = L(\pi_\theta, U) \;-\; \mathbb{E}_{(x,y_w)\sim D}\big[\log \pi_\theta(y_w|x)\big].$$

This BC term brings two concrete benefits:

(i) **Preventing divergence from the true preferred data distribution.** The BC regularizer anchors $\pi_\theta$ to the empirical distribution of preferred samples, preventing the policy from drifting too far away from what is actually observed in the data. This mitigates the risk

of probability mass collapsing onto overly confident or stylistically extreme outputs and stabilizes training, especially in the absence of a strong reference policy.

(ii) **Providing an absolute learning signal beyond relative comparisons.** While the preference loss $L(\pi_\theta, U)$ only tells the model that "$y_w$ is better than $y_l$", the BC term directly encourages high likelihood on $y_w$ itself. This provides an *absolute* supervised signal on preferred outputs, complementing the purely contrastive objective and ensuring that the model learns not only which response is better, but also what a good response should look like in distributional terms.

These properties are particularly important in the medical domain, where the target behavior is highly deterministic and correctness is much more critical than stylistic variation. In such a setting, it is not sufficient to merely prefer one response over another; the model must consistently produce stable, factually accurate outputs that closely match expert-like references. The BC regularizer is therefore especially well-suited here, as it pulls the model toward a sharp, well-calibrated distribution over medically correct responses rather than encouraging diversity or style.

This interpretation is also consistent with our empirical analysis. As shown in Figure 6 of the main text, using a lower sampling temperature leads to better performance. This observation aligns with the role of the BC term: by encouraging higher likelihood on preferred responses, CPO effectively shapes a sharper and more deterministic output distribution, which is desirable in high-stakes medical applications. Together, these theoretical and empirical considerations justify our choice of CPO with BC regularization over standard DPO in the proposed framework.

**GRPO Performance Degradation.** In our fetal ultrasound experiments, we observe that GRPO-based reinforcement learning yields performance degradation compared to supervised models. The root cause is that the conventional rewards optimized by GRPO act only as imperfect proxies for real clinical objectives. Such proxy rewards fail to capture fine-grained anatomical consistency, multi-view joint reasoning, and standardized report structures. The policy therefore tends to overfit these incomplete signals—exploiting phrasing patterns, templates, or other superficial regularities to "game the reward"—while degrading on clinically critical attributes such as localization accuracy, measurement validity, and structural coverage.

Moreover, GRPO updates policies by sampling candidate responses and optimizing based on relative rewards, which introduces additional stochasticity and high-variance gradients. This perturbs the likelihood distribution learned by supervised training—a distribution that is already close to optimal for the near-deterministic mapping required in fetal ultrasound—and drives the policy toward a small set of "reward-seeking" modes, reducing robustness and generalization.

A promising direction is to train a dedicated reward model that evaluates each prediction and provides more clinically aligned feedback, supplying GRPO with learning signals that better reflect real diagnostic criteria. This approach is particularly compelling in complex fetal ultrasound settings that require multi-image reasoning and coverage across diverse anatomical structures and conditions.

### B.4 ANALYSIS OF GPT-BASED VS. DIRECT DIAGNOSIS

For baseline models that do not provide native diagnostic outputs, we adopt a two-step evaluation protocol in which GPT is used to infer diagnostic labels from the generated reports. This indirect procedure can introduce additional noise, since inaccuracies in GPT's second-step extraction may lead to an artificial underestimation of baseline performance. To quantify this effect, we apply the same two-step evaluation to `FetalMind` reports. As shown in the Table 7, the resulting performance degradation is within 1%. Here are several potential sources of error:

- **Misinterpretation by GPT.** GPT may misread the semantics of free-text reports, especially when negations or rhetorical expressions are involved. For instance, "atrial septal defect" can be misclassified as "ventricular septal defect". Our Fetal Token Injection module is explicitly designed to mitigate this issue by introducing special tokens for key anatomical and pathological terms, thereby reducing semantic confusion at the tokenization level.

- **Ambiguous degree modifiers in medical language.** Clinical descriptions frequently include degree-related qualifiers such as "mild" or "suspicious". These modifiers may be

**Table 7:** Analysis of GPT-Based vs. Direct Diagnosis. **Bold** and underlined text indicates the best performance and second-best performance, respectively. Note that * indicates models fine-tuned with *Supervised Fine-Tuning* to ensure a fair comparison.

| Type | Model | #Params | GPT Diagnosis | CE Metrics ↑ | | | ACC↑ | Body F1-20↑ |
|------|-------|---------|---------------|----|----|----|------|-------------|
| | | | | **P** | **R** | **F1** | | |
| **w/ US Train** | Gemini 2.5 Pro | - | ✓ | 19.4 | 16.1 | 17.6 | 71.4 | 26.4 |
| | GPT-5 | - | ✓ | 19.1 | 12.6 | 15.2 | 71.6 | 23.6 |
| | InternVL3 * | 1B | ✗ | 26.2 | 18.9 | 22.0 | 78.2 | 39.9 |
| | FetalMind-S1 | 1B | ✗ | 23.1 | **29.2** | 25.8 | 79.0 | 45.2 |
| | FetalMind-M7 | 7B | ✗ | **34.7** | 28.2 | **31.1** | **81.3** | **50.2** |
| | FetalMind-M7 | 7B | ✓ | 34.2 | 27.6 | 30.7 | 80.6 | 50.0 |

interpreted inconsistently by LLMs under different contexts, leading to over-calling or under-calling certain findings.

- **Heterogeneous reporting styles across centers.** The same underlying condition can be phrased very differently by radiologists at different institutions. For example, Center A might report "Choroid plexus cyst noted in left ventricle", whereas Center B might describe "Anechoic lesion detected within the choroid plexus". Although both correspond to a choroid plexus cyst, the lexical variation can introduce additional challenges for robust automatic parsing.

### B.5 INVESTIGATION OF ERROR SAMPLES

As shown in Fig. Figure 12, we qualitatively analyze representative failure cases of our model on fetal ultrasound. We observe two main error patterns:

- **Over-sensitivity to minor findings.** In small or borderline examinations, the model can be overly sensitive to subtle variations, occasionally assigning pathological labels to findings that experienced clinicians would still consider within normal limits.

- **Inter-center inconsistency in annotation standards.** Different centers may adopt slightly different criteria for the same condition, for example using different thresholds to define increased nuchal translucency (e.g., $> 2.5\,\text{mm}$ vs. $> 2.6\,\text{mm}$). This issue can be mitigated by harmonizing disease labels according to international guidelines and remapping site-specific criteria to a unified standard.

## C TRAINING TEMPLATE

### C.1 FETAL ULTRASOUND REPORT TEMPLATE

To promote both clinical validity and cross-center consistency, we constructed a standardized obstetric ultrasound report template by systematically consolidating and harmonizing recommendations from multiple international guidelines, including those issued by the ISUOG, AIUM, and Chinese Medical Association. As illustrated in Figure 13 and Figure 14, we release both an English and a Chinese version of the template. The English version facilitates alignment with widely adopted global standards, while the Chinese version ensures applicability in large-scale domestic clinical practice. Together, these templates provide a unified and clinically grounded structure for report writing, enabling reliable data annotation, model training, and evaluation. Importantly, by establishing a guideline-based framework, the templates mitigate variability across institutions and languages, offering a scalable foundation for developing deep learning systems that generalize robustly across centers, devices, and populations.

## C.2 Instruction content for clinical expert reference

Below, we present instruction templates for both report generation and diagnostic reasoning. These templates establish a consistent and structured reference framework for clinical experts during model evaluation, ensuring that model outputs are assessed according to unified and standardized criteria.

---

**▣ Diagnosis Template (Expert Reference)**

1. <Aortic arch view><image>2. <Four-chamber view with intracardiac echogenic focus><image>3. <Superior–inferior vena cava view><image>4. <Three-vessel view><image>5. <Four-chamber view><image>6. <Femur long-axis view><image>7. <Biparietal diameter plane><image>8. <Thoracic coronal view><image>. Please provide a professional fetal assessment based on the anatomical ultrasound planes and indicate whether one or more abnormalities are present.

---

**ⓘ Report Template (Expert Reference)**

1. <Biparietal diameter plane><image>2. <Longitudinal view of the spine><image>3. <Cerebellar plane><image>4. <Median sagittal view of the fetal face><image>5. <Lateral ventricle plane><image>6. <Transverse abdominal plane through the bladder><image>7. <Coronal view of both kidneys><image>8. <Interorbital distance plane><image>9. <Transverse view of both kidneys><image>10. <Sagittal view of a single kidney><image>11. <Transverse abdominal diameter measurement plane><image>12. <Three-vessel view><image>13. <Coronal view of the fingers><image>14. <Left ventricular outflow tract view><image>15. <Four-chamber heart view><image>16. <Pulmonary artery bifurcation><image>17. <Coronal view of the lips><image>18. <Gallbladder view><image>19. <Femur long-axis view><image>. Please carefully review the above ultrasound images (the image order and anatomical regions have been specified). After an overall assessment, please provide a detailed description of the Findings and Impression.

---

# D EVALUATION METRICS

In this section, we provide a detailed mathematical formulation of common metrics used for evaluating Natural Language Generation (NLG) tasks and Classification Evaluation (CE) tasks. These metrics, such as BLEU, METEOR, ROUGE-L, Precision, Recall, and F1-Score, are used to assess the quality and effectiveness of machine-generated text in comparison to ground truth references.

## D.1 BLEU (B-1 AND B-4)

BLEU (Bilingual Evaluation Understudy) measures the precision of n-grams between the generated and reference texts. It is often used for machine translation and other NLG tasks. BLEU considers the precision of unigrams (B-1) and 4-grams (B-4), calculating the overlap between the generated text and reference texts.

$$\text{B-1} = \text{Precision}_1 = \frac{\sum_{n=1}^{N} \text{Count}_{\text{match},1}}{\sum_{n=1}^{N} \text{Count}_{\text{generated},1}} \tag{10}$$

$$\text{B-4} = \text{Precision}_4 = \frac{\sum_{n=1}^{N} \text{Count}_{\text{match},4}}{\sum_{n=1}^{N} \text{Count}_{\text{generated},4}} \tag{11}$$

Where: - $\text{Count}_{\text{match},n}$ represents the number of n-grams that appear in both the reference and the generated text. - $\text{Count}_{\text{generated},n}$ represents the total number of n-grams in the generated text.

BLEU can be extended with a brevity penalty (BP) to account for the length of the generated text:

$$\text{BLEU} = \text{BP} \times \exp\left(\sum_{n=1}^{N} w_n \log p_n\right) \tag{12}$$

Where $w_n$ is the weight for each n-gram (usually uniform), and $p_n$ is the precision of n-grams of size $n$.

## D.2 METEOR (MTR)

METEOR (Metric for Evaluation of Translation with Explicit ORdering) improves upon BLEU by incorporating synonymy, stemming, and word-order preservation. METEOR balances precision and recall with an F-score, considering the meaning of words (synonyms) and morphological variations (stemming).

$$\text{MTR} = \text{F(Precision, Recall, Synonymy, Stemming)} \tag{13}$$

Where: - Precision is the proportion of generated words that match the reference words. - Recall is the proportion of reference words that match the generated words. - Synonymy adjusts for synonyms (i.e., different words with similar meanings). - Stemming adjusts for different forms of the same word (e.g., "running" vs. "run").

The F-measure is used to combine precision and recall:

$$\text{F(P, R)} = \frac{10 \cdot \text{P} \cdot \text{R}}{9 \cdot \text{P} + \text{R}} \tag{14}$$

## D.3 ROUGE-L (R-L)

ROUGE (Recall-Oriented Understudy for Gisting Evaluation) is a set of metrics primarily used for evaluating machine-generated summaries. The ROUGE-L metric focuses on the longest common subsequence (LCS) between the reference and generated text, which captures the order of the words.

The ROUGE-L score is calculated as:

$$\text{R-L} = \frac{LCS(\text{generated, reference})}{\text{Length of reference}} \tag{15}$$

Where $LCS(\text{generated, reference})$ is the length of the longest common subsequence between the generated text and the reference text. The LCS metric encourages the preservation of word order, which is crucial for the quality of text generation.

Additionally, ROUGE can be extended to compute recall ($R$) and precision ($P$) as follows:

$$R = \frac{\text{LCS}}{\text{Length of reference}}, \quad P = \frac{\text{LCS}}{\text{Length of generated text}} \tag{16}$$

## D.4 PRECISION (P)

Precision is a metric used in classification tasks, which measures the accuracy of the predictions by comparing the true positives (TP) to the total predicted positives (TP + FP):

$$P = \frac{\text{TP}}{\text{TP} + \text{FP}} \tag{17}$$

Where: - TP represents the number of true positive instances (correctly predicted relevant instances). - FP represents the number of false positive instances (incorrectly predicted relevant instances).

### D.5   RECALL (R)

Recall measures how well the classifier identifies all relevant instances by comparing the true positives (TP) to the total number of actual positives (TP + FN):

$$R = \frac{\text{TP}}{\text{TP} + \text{FN}} \tag{18}$$

Where: - FN represents the number of false negative instances (relevant instances that were incorrectly predicted as irrelevant).

### D.6   F1 SCORE (F1)

The F1 Score is a harmonic mean of precision and recall, providing a balanced measure of classification performance. It is particularly useful when dealing with imbalanced datasets:

$$F1 = 2 \times \frac{P \times R}{P + R} \tag{19}$$

The F1 Score is maximized when both precision and recall are high, making it an excellent metric when both false positives and false negatives are equally important.

### D.7   MACRO AND MICRO AVERAGING FOR PRECISION, RECALL, AND F1

In multi-class classification tasks, we often calculate macro and micro averages for precision, recall, and F1 score:

**Macro Average:** The macro average treats all classes equally by averaging the individual scores of each class:

$$\text{Macro } P = \frac{1}{C}\sum_{i=1}^{C} P_i, \quad \text{Macro } R = \frac{1}{C}\sum_{i=1}^{C} R_i, \quad \text{Macro } F1 = \frac{1}{C}\sum_{i=1}^{C} F1_i \tag{20}$$

Where $C$ is the number of classes, and $P_i$, $R_i$, and $F1_i$ are the precision, recall, and F1 scores for class $i$.

**Micro Average:** The micro average aggregates the true positives, false positives, and false negatives across all classes and then calculates the precision, recall, and F1:

$$\text{Micro } P = \frac{\sum_{i=1}^{C} \text{TP}_i}{\sum_{i=1}^{C}(\text{TP}_i + \text{FP}_i)}, \text{Micro } R = \frac{\sum_{i=1}^{C} \text{TP}_i}{\sum_{i=1}^{C}(\text{TP}_i + \text{FN}_i)},$$
$$\text{Micro } F1 = 2 \times \frac{\text{Micro } P \times \text{Micro } R}{\text{Micro } P + \text{Micro } R} \tag{21}$$

Where $\text{TP}_i$, $\text{FP}_i$, and $\text{FN}_i$ are the true positives, false positives, and false negatives for class $i$, respectively.

## E   THE USE OF LARGE LANGUAGE MODELS (LLMS)

During manuscript preparation, we employed large language models (LLMs), specifically GPT-5, strictly as writing assistants to enhance grammar, clarity, and readability. Their role was limited to rephrasing for improved flow and correcting typographical errors. The scientific ideas, experimental design, analyses, and conclusions were conceived and developed entirely by the human authors. All model-generated text was carefully reviewed and edited by the authors, who take full responsibility for the manuscript's accuracy and originality.

**Case #3329**

**User Query:** 1. \<image\> 2. \<image\> 3. \<image\> 4. \<image\> 5. \<image\>. Please review each image in sequence (anatomical site annotated) and, based on all the images, summarize the FINDINGS and IMPRESSION.

**Ground Truth**

FINDINGS:
Ultrasound description:
Fetal heart and movements: visible; cephalic presentation.
Anatomical structures: spine visualized.

IMPRESSION:          **Diagnosis : Normal**
Spine re-examination: visualized.

**FetalMind**

FINDINGS:
Ultrasound description:
Fetal heart and movements: visible.
Left renal pelvis: 7.5 mm; right renal pelvis: 10.1 mm.
          **Diagnosis: Bilateral renal pelvic dilatation**
IMPRESSION:
Bilateral renal pelvic dilatation.

**Case #34212**

**User Query:** \<image\> 2. \<image\> 3. \<image\> 4. \<image\> 5. \<image\> 6. \<image\> 7. \<image\> 8. \<image\> 9. \<image\> 10. \<image\> 11. \<image\> 12. \<image\> 13. \<image\> 14. \<image\> 15. \<image\> 16. \<image\>.. Please review each image in sequence (anatomical site annotated) and, based on all the images, summarize the FINDINGS and IMPRESSION.

**Ground Truth**

FINDINGS: Ultrasound description:
[Transabdominal] Number of fetuses: 1; Fetal position: cephalic; Fetal heart activity and movements: present.
[Fetal biometric measurements]
Crown–rump length (CRL): 73 mm;
Biparietal diameter: 23 mm;
Head circumference: 86 mm;
Abdominal circumference: 65 mm;
Femur length: 9 mm;          **Diagnosis : Normal**
Humerus length: 8 mm.
Placental location: anterior wall and right lateral wall; placental thickness: 15 mm; placental maturity: grade 0; maximum amniotic fluid pocket depth: 32 mm.
[Nuchal translucency (NT) thickness]: 2.6 mm.
The following anatomical structures are visualized:
Head: cranial halo; midline of the brain; "butterfly sign"; left orbit; right orbit.
Chest: thoracic cavity.
Abdomen: abdominal wall; gastric bubble; bladder; umbilical cord vessels: 3.
Limbs: left femur; left tibia and fibula; right femur; right tibia and fibula; left humerus; left radius and ulna; left wrist; right humerus; right radius and ulna; right wrist.
[M-mode ultrasound] Fetal heart rate: 157 beats per minute.
[Uterine arteries] Uterus appears in a pregnant state.
On the medial side of the right pelvic wall, the right uterine artery runs upward and forward, crossing the right external iliac artery; PI: 0.64; RI: 0.46; early diastolic notch: not seen.
On the anteromedial side of the left pelvic wall, the left uterine artery runs upward and forward, crossing the left external iliac artery; PI: 1.82; RI: 0.77; early diastolic notch: not seen.
IMPRESSION: Singleton, viable fetus. Fetal size corresponds to 13 weeks 3 days of gestation.

**FetalMind**

FINDINGS: Ultrasound description:
[Transabdominal] Number of fetuses: 1; Fetal position: cephalic; Fetal heart activity and movements: present.
[Fetal biometric measurements]
Crown–rump length (CRL): 73 mm;
Biparietal diameter: 23 mm;
Head circumference: 86 mm;
Abdominal circumference: 66 mm;
Femur length: 8 mm;          **Diagnosis: Increased nuchal translucency**
Humerus length: 8 mm.
Placental location: anterior wall; placental thickness: 13 mm; placental maturity: grade 0; maximum amniotic fluid pocket depth: 31 mm.
[Nuchal translucency (NT) thickness]: 2.6 mm.
The following anatomical structures are visualized:
Head: cranial halo; midline of the brain; "butterfly sign"; left orbit; right orbit.
Chest: thoracic cavity.
Abdomen: abdominal wall; gastric bubble; bladder; umbilical cord vessels: 3.
Limbs: left femur; left tibia and fibula; right femur; right tibia and fibula; left humerus; left radius and ulna; left wrist; right humerus; right radius and ulna; right wrist.
[M-mode ultrasound] Fetal heart rate: 159 beats per minute.
[Uterine arteries] Uterus appears in a pregnant state.
On the medial side of the right pelvic wall, the right uterine artery runs upward and forward, crossing the right external iliac artery; PI: 1.31; RI: 0.69; early diastolic notch: not seen.
On the anteromedial side of the left pelvic wall, the left uterine artery runs upward and forward, crossing the left external iliac artery; PI: 1.32; RI: 0.66; early diastolic notch: not seen.
IMPRESSION: Singleton pregnancy. Fetal size corresponds to 13 weeks 3 days of gestation. Increased nuchal translucency (NT)

**Figure 12:** Illustration of `FetalMind` error samples identified during evaluation.

**Figure 13:** The generalized version of our obstetric ultrasound report template, established with reference to multiple international clinical guidelines. It provides a consistent and clinically grounded format for training and evaluating deep learning systems.

**Figure 14:** The Chinese version of our obstetric ultrasound report template, established with reference to multiple international clinical guidelines. It provides a consistent and clinically grounded format for training and evaluating deep learning systems.

