# OpenReview forum: "Epistemic-Aware Vision–Language Foundation Model for Fetal Ultrasound Interpretation"
_ICLR.cc/2026/Conference — ICLR 2026 Conference Withdrawn Submission_

### Official Review · Reviewer_bzL1 · 2025-10-20

**Soundness:** 3
**Presentation:** 3
**Contribution:** 3
**Rating:** 6
**Confidence:** 4

**Summary:**

This paper introduces FetalMind, a medical MLLM designed for diagnosis and report generation from fetal ultrasound images. The authors show that existing medical MLLMs and general purpose models perform poorly on complex fetal scans, while FetalMind incorporates a new approach called Salient Epistemic Disentanglement, which helps the model understand relationships between ultrasound views and diseases using a bipartite knowledge graph. The authors also present FetalSigma-1M, a large dataset of fetal ultrasound images and 20k clinical reports from 12 medical centers. Experiments show that FetalMind outperforms both medical MLLMs and general purpose models for diagnosing fetal ultrasound scans in various gestational stages.

**Strengths:**

- They introduce FetalSigma-1M, a large and diverse dataset covering all trimesters and multiple ultrasound views, which can help the community further advance diagnosis of fetal ultrasound scans.
- Their contribution like SED and Spatial alignment help the model outperform general purpose MLLMs validating domain aware contributions.

**Weaknesses:**

- Concatenating disease related salient view from one fetus to another one is not very indicative of real world scenario. There could be unique characteristics of the receiver’s fetus which does not correspond with the donor’s disease related images. It might help improve the limited benchmarks but could fail in real clinical cases.
- It is unclear what level of effort is required to create and verify the bipartite knowledge graph, and whether this process can easily scale to new disease types.
- During the SVPO training the model might get biased towards the positive disease related salient views since they are concatenated with other normal views. In such a case, does the MLLM even require non salient views, can the model reach a similar accuracy by only using the salient views from the knowledge graph?

**Questions:**

Please address the weaknesses above
- Add examples of bipartite knowledge graph and its clinical verification process in appendix.
- What instructions are given to clinical experts for diagnosis verification?
- Line 105: with 1B and 7b versions. 7B should be consistent.

---

> ### Author Response · Authors · 2025-11-18
> **Authors Rebuttal to Reviewer bzL1 (1/2)**
>
> We sincerely thank you for your thorough feedback: the paper is clearly written, the proposed method is well-motivated, easy to reimplement, and the experiments are well organized. We address your questions below and have incorporated all feedback in the revised version. **All new material added to the revised manuscript has been highlighted in blue text for better visibility.**
>
> > 1. Concatenating disease related salient view from one fetus to another one is not very indicative of real world scenario. There could be unique characteristics of the receiver’s fetus which does not correspond with the donor’s disease related images. It might help improve the limited benchmarks but could fail in real clinical cases.
>
> A1: Thank you for your insightful suggestion. It is important to note that the sample replacement part of our SED is only used within the diagnostic process and does not involve report generation. Our design is based on a clinical observation: "The decisive role of pathology views in diagnosis." In clinical practice, a doctor can diagnose a fetal condition if an abnormal pathology is identified in any of the standard views and remains present after re-examination.
>
> From a perturbation perspective, although individual fetuses may have unique physiological features, their normal characteristics remain consistent, leading to negative diagnostic results. The inclusion of "donor" pathology images, however, could evolve into new diseases. The performance improvement further supports the validity of our approach.
>
> To further validate the effectiveness of our method, we conducted a real-world clinical scenarios test on 56 cases from two centers, as shown in Table. Specifically, we set up three groups: one with a moderately experienced doctor, one with a moderately experienced doctor + AI, and a true clinical control group (comprising three doctors, including one with higher experience). Diagnoses were conducted simultaneously after image acquisition. As shown in the table below, FetalMind was able to assist doctors in improving report structure and enhancing diagnostic accuracy.
>
> We have included this analysis in the revised version in Appendix A.6: REAL-WORLD CLINICAL DECISION-MAKING ANALYSIS.
>
> **Overall comparison of NLP and classification metrics between Doctor and Doctor+AI**
>
> | Method     | BLEU-1  | BLEU-4  | ROUGE-1 | ROUGE-L | METEOR | Precision_micro | F1_micro |
> |---|----|-|----|-----|-----|----|-----|
> | Doctor     | 75.388  | 67.817  | 77.450  | 72.019  | 30.592 | 0.568           | 0.562    |
> | Doctor+AI  | **88.532** | **81.605** | **86.002** | **85.717** | **59.351** | **0.679** | **0.653** |
>
> ---
>
> > 2. It is unclear what level of effort is required to create and verify the bipartite knowledge graph, and whether this process can easily scale to new disease types.
>
>
> A2: We appreciate the reviewer for raising this point, as it indeed reflects substantial human effort. Specifically, the graph was constructed by five clinicians, and the entire process spanned approximately four weeks, accumulating around 40–60 expert person-hours. This investment highlights the level of expertise and manual workload required to build a high-quality medical knowledge graph.
>
> Our disease–view bipartite graph contains 326 disease nodes, 54 view nodes, and 879 edges. All nodes were derived from textbooks, clinical guidelines, and expert consensus, and further standardized using unified terminology to ensure consistency. We also clarify the expert-in-the-loop procedure: three clinicians with over 10 years of experience reviewed the preliminary disease–view relations, refined them, and conducted multiple discussion rounds. When disagreements emerged, we adopted a Delphi-style anonymous voting strategy or arbitration by a senior third expert, as explained in the revised manuscript.
>
> In the revised version, we visualize the disease–view bipartite graph in A.7 VISUALIZATION OF THE DISEASE–VIEW GRAPH. Upon acceptance, we will release the complete graph data for transparency and reproducibility.
>
> **New Disease Types** Your suggestion is highly insightful. For new disease categories or subtypes, the bidirectional mapping enabled by our knowledge graph can be used to extend FetalMind to additional conditions with minimal finetuning cost. We note that one major challenge in fetal ultrasound is the time-consuming reporting process and the relatively low diagnostic efficiency, both of which FetalMind is designed to improve. We will further discuss the extension to new disease types in future work.

---

> ### Author Response · Authors · 2025-11-18
> **Authors Rebuttal to Reviewer bzL1 (2/2)**
>
> > 3. During the SVPO training the model might get biased towards the positive disease related salient views since they are concatenated with other normal views. In such a case, does the MLLM even require non salient views, can the model reach a similar accuracy by only using the salient views from the knowledge graph?
>
> A3. Thank you for the insightful comment! This concern appears to stem from a mismatch. *If not, we would greatly appreciate your clarification.* For a newly diagnosed fetus, a complete ultrasound examination naturally consists of multiple images that include both salient and non-salient views, and these cannot be distinguished beforehand. Salient views are not predefined; they are identified after the model produces a diagnosis, by mapping the predicted disease to its corresponding views through the knowledge graph. This occurs post-decision and does not assume any prior knowledge during inference.
>
> During the reinforcement learning stage, however, we leverage the knowledge graph to map the ground-truth diagnosis to its associated views in order to distinguish salient from non-salient ones. This is intentional and aligns with the purpose of SED: to inject structured disease–view knowledge, strengthen the association between diseases and views, and encourage the model to attend more effectively to disease-related regions.
>
>
> > Q1. Add examples of bipartite knowledge graph and its clinical verification process in appendix.
>
> A4: Thank you for your valuable insights. In the revised version, we have added an example of the bipartite knowledge graph in Appendix A.7 **Visualization of the Disease–View Graph**. The clinical validation process has been provided as referenced in our previous response.
>
> ---
>
> > Q2. What instructions are given to clinical experts for diagnosis verification?
>
> A5: Thanks. We provide instructions for both report generation and diagnosis in Appendix C.2: **Instruction Content for Clinical Expert Reference**.
>
> ---
>
> > Q3. Line 105: with 1B and 7b versions. 7B should be consistent.
>
> A6: Thank you for pointing this out. We have corrected the inconsistency and updated **1B and 7b versions** to **1B and 7B versions** in Line 105.

---

> ### Public Comment · ~He_Xiao1 · 2026-03-01
>
> ***Dear Reviewer bzL1,***
>
> I hope this message finds you well.
>
> I am one of the authors of Paper 1306, titled "Epistemic-Aware Vision–Language Foundation Model for Fetal Ultrasound Interpretation." We noticed that you did not participate in the post-rebuttal discussion, and we wanted to kindly ask whether your final assessment remains consistent with your initial positive review.
>
> Could you please confirm if your score still stands, or if you were considering raising it?
>
> Thank you very much for your time and consideration.
>
> Best regards,
>
> Xiao He, Huangxuan Zhao, Guojia Wan, Zhou Wei, Yanxing Liu, Juhua Liu, Yongchao Xu, Yong Luo, Dacheng Tao, Bo Du (Wuhan University & Nanyang Technological University)

---

### Official Review · Reviewer_tEu6 · 2025-10-22

**Soundness:** 2
**Presentation:** 2
**Contribution:** 2
**Rating:** 4
**Confidence:** 3

**Summary:**

This paper introduces FetalMind, a vision-language medical model tailored for fetal ultrasound report generation and diagnosis. The core contributions include a novel Salient Epistemic Disentanglement (SED) strategy that explicitly injects a bipartite disease-view knowledge graph into the model, combined with a preference optimization method (SVPO) using reinforcement learning to decouple disease-view associations in a clinically faithful manner. The model is trained on the FetalSigma-1M dataset, comprising over 1 million ultrasound images and 20,000 reports from 12 medical centers, which addresses the data scarcity in fetal ultrasound AI. Extensive multi-center experiments show strong gains over state-of-the-art medical and generalist multimodal LLMs, and include detailed ablations and visualizations.

**Strengths:**

1. The paper addresses the core challenges in fetal ultrasound analysis areas such as multi-view reasoning and disease diversity where existing models generally perform suboptimally. FetalMind is designed to tackle a genuine and pressing clinical need.
2.SED integrates expert prior knowledge (via bipartite graphs), powerful representation learning capabilities, and sophisticated alignment techniques (SVPO reinforcement learning). This workflow-mimicking design not only improves performance but also enhances the interpretability and reliability of the model’s reasoning process.
3. The introduction of the FetalSigma-1M dataset is a large-scale, multi-center, multi-view corpus in fetal ultrasound with fine-grained disease labels and standardized reports. This is likely to have enduring value for the community.
4. Extensive evaluation across gestational ages and multiple sites/devices establishes potential for generalization and robustness. Ablation studies are performed in depth, quantifying the impact of each component.

**Weaknesses:**

1. While the related work section describes several prior MLLMs and medical VLMs, there is insufficient comparison (both qualitative and, where possible, quantitative) with very recent medical vision-language foundation models, such as FetalCLIP (F Maani et al., 2025), EchoCLIP (M Christensen et al., 2024), and VividMed (L Luo et al., 2024).
2. The paper presents SVPO as one of its contributions, yet its core loss function is identical to that of CPO (Xu et al., 2024). Compared with DPO, which removes the regularization supervision from the reference model, the ablation study does not directly provide a fair comparison between SVPO and DPO trained on the same data, thereby failing to isolate and validate the claimed advantage of "no reference model." Additionally, the reasons why training with DPO yields results inferior to Vanilla require further explanation.
3. The SVPO method generates training samples by switching significant views between different cases. While this represents an innovative data augmentation approach, it may also introduce unnatural or even anatomically implausible combinations. For example, it could combine a cardiac view from a 34-week fetus with an abdominal view from a 22-week fetus. The paper does not adequately address the potential negative impacts of such synthetic data, nor does it discuss how the model differentiates these artificial cases from real-world multi-lesion cases that, while complex, adhere to physiological logic. This raises concerns that the model might learn shortcuts for identifying splicing artifacts rather than acquiring genuine clinical reasoning abilities.
4. While strong positive examples are given, there is little explicit discussion of where the proposed approach fails or underperforms, nor are there error analyses.

**Questions:**

1. The expert-curated "disease-view" bipartite graph is central to the SED methodology. While the paper notes that this graph was constructed under the guidance of textbooks and experts, it lacks further specific details. For instance, what is the scale of this graph? How many disease and view nodes are included? How are discrepancies in expert opinions handled? Supplementing these details would enhance the reproducibility and transparency of the approach.
2. The paper proposed Fetal Token Injection to improve the model's ability to distinguish between clinically distinct but semantically similar terms. Could you provide more technical details on its implementation? For example, are these new tokens added to the language model's tokenizer? How are their embeddings initialized and trained? How does this mechanism technically enforce the clear separability you claim, beyond standard fine-tuning?

---

> ### Author Response · Authors · 2025-11-18
> **Authors Rebuttal to Reviewer tEu6 (1/4)**
>
> We sincerely thank you for your thorough feedback: the paper is clearly written, the proposed method is well-motivated, easy to reimplement, and the experiments are well organized. We address your questions below and have incorporated all feedback in the revised version. **All new material added to the revised manuscript has been highlighted in blue text for better visibility.**
>
> > 1. While the related work section describes several prior MLLMs and medical VLMs, there is insufficient comparison (both qualitative and, where possible, quantitative) with very recent medical vision-language foundation models, such as FetalCLIP (F Maani et al., 2025), EchoCLIP (M Christensen et al., 2024), and VividMed (L Luo et al., 2024).
>
> A1: We thank the reviewers for pointing out related works such as **FetalCLIP**, **EchoCLIP**, and **VividMed**. Below we clarify how these methods relate to our setting and why a direct, fair comparison at the level of *full fetal diagnosis and report generation* is not feasible.
>
> **FetalCLIP.** We have successfully run FetalCLIP in our environment and already cited it in the main paper (L560). However, FetalCLIP is not directly comparable to FetalMind in terms of end-to-end diagnostic capability. Specifically, FetalCLIP focuses on four computer vision tasks: 1. Standard view classification for fetal ultrasound images; 2. Gestational age estimation; 3. Congenital heart disease (CHD) detection; 4. Segmentation of fetal anatomical structures across different views.
>
> The outputs of FetalCLIP are feature embeddings and conventional CV predictions (classification, detection, segmentation), rather than structured diagnostic reports or clinically actionable conclusions. In other words, FetalCLIP operates at the *single-plane image* level and does not integrate information across multiple views to produce a comprehensive fetal diagnosis. By contrast, the goal of FetalMind is to build a system that can **assist clinicians in multi-view, multi-disease decision-making**, going beyond image-level recognition or classification to generate interpretable diagnostic reports.
>
> **EchoCLIP.** EchoCLIP is primarily developed for **adult or neonatal echocardiography**, not for **fetal echocardiography**. Furthermore, fetal cardiac abnormalities represent only one subset of overall fetal abnormalities and do not cover the full spectrum of fetal conditions modeled in FetalMind. As a result, EchoCLIP does not provide a directly comparable benchmark for our *whole-fetus, multi-organ, multi-condition* setting.
>
> **VividMed.** We carefully reviewed VividMed and found that it does **not** perform pretraining on ultrasound data, nor does the paper mention ultrasound-specific experiments. In addition, the official repository (see the MMMM project page) does not provide a clear, end-to-end reproduction pipeline suitable for our fetal ultrasound setup.
>
>
> **Breadth of comparison.** In our experiments, we also observed that many state-of-the-art medical VLMs either (i) do not support multi-image input, or (ii) even when extended to multi-image input, underperform strong general-purpose models (as shown in Table 1 of the main paper). Therefore, we prioritized reproducing **the best-performing and practically usable baselines**, with special focus on medically pretrained models and strong general models such as **Gemini Pro 2.5**.
>
> It is worth noting that VividMed compares against 6 methods, and FetalCLIP against 3 methods, while our FetalMind evaluation includes **9 baselines** and additionally fine-tunes both a medical VLM (LLaVA-Med) and a general VLM. This results in a broader and stronger set of comparisons in our targeted setting of multi-image, multi-finding fetal ultrasound reasoning.

---

> ### Author Response · Authors · 2025-11-18
> **Authors Rebuttal to Reviewer tEu6 (2/4)**
>
> > 2. The paper presents SVPO as one of its contributions, yet its core loss function is identical to that of CPO (Xu et al., 2024). Compared with DPO, which removes the regularization supervision from the reference model, the ablation study does not directly provide a fair comparison between SVPO and DPO trained on the same data, thereby failing to isolate and validate the claimed advantage of "no reference model." Additionally, the reasons why training with DPO yields results inferior to Vanilla require further explanation..
>
> A2: Thank you for your valuable feedback. In our ablation experiments, the reinforcement learning training data for DPO and CPO are strictly identical. Both methods are trained on the same set of preference pairs (positive–negative sample pairs), ensuring a fair comparison under matched data and supervision formats. Our core contribution does not lie in proposing a novel loss function per se, but rather in the representation learning strategy and the multi-image training sample generation mechanism (in line with the spirit of MIA-DPO). If the reviewer has more specific evidence or scenarios where the comparison may be unfair, we would sincerely appreciate further clarification.
>
> The performance of DPO remains notably worse than the vanilla model. We attribute this mainly to the fact that DPO relies on a fixed reference policy $\pi_{\text{ref}}$ (typically an SFT model) and optimizes $L(\pi_\theta; \pi_{\text{ref}})$, implicitly constraining $\pi_\theta$ to stay close to a reference that often underfits domain-specific medical knowledge. This can cause the model to overfit superficial stylistic cues in the preference data (e.g., verbosity, confident tone), while harming factual correctness.
>
> CPO removes dependence on $\pi_{\text{ref}}$ by adopting a uniform reference and adding a BC regularizer: ${E}_{(x,y_w)\sim D}.$ The BC term (i) prevents divergence from the true preferred data distribution and (ii) provides an absolute supervised signal on $y_w$, beyond relative “win–lose’’ comparisons. These properties are particularly important in medical applications, where behavior is highly deterministic and correctness is far more crucial than style. Empirically, reducing sampling temperature improves performance, consistent with CPO+BC shaping a sharper and more deterministic output distribution, which further justifies our choice of CPO with BC regularization over standard DPO.
>
> ---
>
> > 3-1. The SVPO method generates training samples by switching significant views between different cases. While this represents an innovative data augmentation approach, it may also introduce unnatural or even anatomically implausible combinations. For example, it could combine a cardiac view from a 34-week fetus with an abdominal view from a 22-week fetus.
> A3-1: Your comments are very insightful. Early- and mid-to-late–pregnancy stages are swapped independently to account for their morphological differences. The key imaging features of most fetal structural abnormalities remain highly stable from the mid to late gestational period. Structural markers used to assess cardiac or abdominal anomalies, such as septal continuity, cardiac axis deviation, and abdominal wall integrity, show consistent diagnostic appearances on standard views obtained at both 22 and 34 weeks of gestation.
>
> We appreciate your thoughtful comment! From a perturbation perspective, different fetuses may exhibit individual physiological variations, but their normal structures remain consistently negative from a diagnostic standpoint; only the introduction of pathological “donor” views changes the diagnostic label. The performance gains we observe empirically further support the effectiveness of this design.
>
> It is important to note that the sample replacement mechanism in our SVPO module is exclusively applied within the diagnostic pipeline and is never used during report generation. The intention is to enhance diagnostic sensitivity and specificity by (i) directing the model’s focus toward disease-bearing views and (ii) injecting graph-based prior knowledge to reinforce the associations both among diseases and between diseases and images.
>
> We have added a detailed explanation of this in Section 4.3 *Salient Epistemic Disentanglement* of the revised manuscript.

---

> ### Author Response · Authors · 2025-11-18
> **Authors Rebuttal to Reviewer tEu6 (3/4)**
>
> A3-2: On the other hand, we consider SVPO not merely as a data augmentation technique, but also as a mechanism for injecting graph-based knowledge to enhance the associations between diseases and between diseases and images, in line with the binary knowledge graph emphasis raised by Reviewer 4 (bzL1). In addition, the multi-image reinforcement learning strategy further improves the discriminability of different pathological conditions.
>
> Finally, to further validate the practical utility of our method, we conducted a real-world scenarios study on 56 cases from two centers, as shown in the table below. We considered three groups: (1) a mid-level clinician alone, (2) the same mid-level clinician assisted by our AI system, and (3) a real clinical control group consisting of three clinicians (including at least one senior expert). As shown in Table 1, FetalMind effectively assists clinicians, improving both the structural quality of the written reports and the diagnostic accuracy. We have added this analysis to Appendix A.6 *REAL-WORLD CLINICAL DECISION-MAKING ANALYSIS* in the revised manuscript.
>
> **Overall comparison of NLP and classification metrics between Doctor and Doctor+AI**
> | Method     | BLEU-1 | BLEU-4 | ROUGE-1 | ROUGE-L | METEOR | Precision_micro | F1_micro |
> |-----|--|-|---|---|--|---|--|
> | Doctor | 75.388 | 67.817 | 77.450  | 72.019  | 30.592 | 0.568 | 0.562   |
> | Doctor+AI | **88.532** | **81.605** | **86.002** | **85.717** | **59.351** | **0.679**         | **0.653** |
>
> ---
>
>
> > 3-2. The paper does not adequately address the potential negative impacts of such synthetic data, nor does it discuss how the model differentiates these artificial cases from real-world multi-lesion cases that, while complex, adhere to physiological logic. This raises concerns that the model might learn shortcuts for identifying splicing artifacts rather than acquiring genuine clinical reasoning abilities.
>
> A3-3: Valuable feedback! To demonstrate that disease recognition does not arise from view bias, we visualize the disease–view graph in Appendix A.7 *Visualization of the Disease–View Graph*. As shown there, a single four-chamber cardiac view can correspond to more than ten distinct fetal cardiac abnormalities, and the alignment between disease–view relations is notably high. This makes it difficult to infer diseases solely from the combination of salient views, supporting that our model does not rely on view bias for disease identification.
>
> A key pain point for existing multimodal large models in multi-image settings is that they are often unable to reliably identify **which** images actually contain pathological findings. However, clinical reasoning in fetal ultrasound necessarily proceeds in two steps: (i) recognizing which views are pathological, and then (ii) reasoning over these pathological views to infer the underlying conditions. Our design is explicitly motivated by the clinical observation that *disease-specific views play a decisive role in diagnosis*.
>
> ----
>
> > 4. While strong positive examples are given, there is little explicit discussion of where the proposed approach fails or underperforms, nor are there error analyses.
>
> A4: Thank you for your insightful suggestion. We briefly discussed the limitations of our method around line 471 in the main paper. In response to the reviewer’s suggestions, we have further expanded Section B.5 INVESTIGATION OF ERROR SAMPLES in the revised version, adding more detailed case descriptions and analysis of typical failure modes.
>
> ---
>
> > Q1. The expert-curated "disease-view" bipartite graph is central to the SED methodology. While the paper notes that this graph was constructed under the guidance of textbooks and experts, it lacks further specific details. For instance, what is the scale of this graph? How many disease and view nodes are included? How are discrepancies in expert opinions handled? Supplementing these details would enhance the reproducibility and transparency of the approach.
>
> A5: Thank you for raising this point! Our disease–view bipartite graph contains 326 disease nodes, 54 view nodes, and 879 edges. All nodes are defined based on standard textbooks, clinical guidelines, and expert consensus, and the terminology is further normalized to ensure consistency. We also provide more details on the expert-in-the-loop construction process: three clinicians with over 10 years of experience independently reviewed and refined the initial disease–view mappings, including supplementation and multiple rounds of discussion. In cases of disagreement, we adopted a Delphi-style anonymous voting procedure or arbitration by a third senior expert, as clarified in the revised manuscript.
>
> In the revised version, we visualize the disease–view bipartite graph in Section A.7 VISUALIZATION OF THE DISEASE–VIEW GRAPH. After acceptance, we will release the complete disease–view graph to facilitate future research.

---

> ### Author Response · Authors · 2025-11-18
> **Authors Rebuttal to Reviewer tEu6 (4/4)**
>
> > Q2. The paper proposed Fetal Token Injection to improve the model's ability to distinguish between clinically distinct but semantically similar terms. Could you provide more technical details on its implementation? For example, are these new tokens added to the language model's tokenizer? How are their embeddings initialized and trained? How does this mechanism technically enforce the clear separability you claim, beyond standard fine-tuning?
>
> A6: Thanks for your comment! As described in the caption of Figure 3, the “disease–view keywords as **special tokens**” mechanism explicitly maps each clinically meaningful disease and view keyword to an independent token. The model then learns dedicated embeddings for these special tokens, which helps avoid errors caused by semantic similarity at the surface-text level.
>
> For example, “atrial septal defect” and “ventricular septal defect” are semantically similar and both may appear in fetal cardiac ultrasound images, but they correspond to different disease entities and often to different diagnostic findings and views. Under standard fine-tuning, these terms may be decomposed into overlapping subword fragments (e.g., “septal” and “defect”), causing the model to treat them as highly similar and making it difficult to capture the clinically critical distinctions between them. In contrast, under our “fetal token injection” design, “atrial septal defect” and “ventricular septal defect” are introduced as separate special tokens. This has two main advantages:
>
> - **Independent tokens.** All diseases and view keywords (e.g., “left ventricular outflow tract”, “aortic coarctation”) is added to the model as an independent special token. This guarantees that every term has its own unique representation and will not be split into subwords or conflated with other terms.
>
> -	**Explicit separation in embedding space.**  By explicitly defining and learning embeddings for these special tokens, the model can form well-separated regions in the embedding space for different disease–view concepts. This enables the model to better distinguish terms that are semantically similar in natural language but correspond to distinct clinical entities in fetal ultrasound.

---

> ### Comment · Reviewer_tEu6 · 2025-11-26
>
> Thank you for the detailed rebuttal. The new experiments and clarifications have resolved most of my initial concerns and significantly improved the paper.
>
> I have three final suggestions to further enhance the manuscript:
>
> 1. Qualitative Comparison: While the quantitative comparison with FetalCLIP is justifiably omitted, the Related Work section would benefit from a deeper qualitative discussion. Positioning FetalMind vs. FetalCLIP within the clinical workflow and discussing their potential synergy would better contextualize your contribution.
>
> 2. Clarity on "SVPO": The name "SVPO" could be misleading. I suggest explicitly stating in the main text that the novelty lies in the "Salient View" data strategy for constructing preference pairs, not in a new "Preference Optimization" algorithm. This will prevent reader confusion.
>
> 3. Shortcut Learning Risk: The new real-world clinical evaluation is a strong addition. However, the theoretical risk of the model learning "splicing artifacts" from the synthetic data remains. Please briefly and candidly discuss this in the Limitations section to enhance the paper's rigor.

---

> ### Author Response · Authors · 2025-11-26
> **Authors Discussion to Reviewer tEu6**
>
> Thank you for recognizing the substantial effort we invested in incorporating evaluations from real clinical settings.
>
> We appreciate your engagement in the discussion, which has helped improve the clarity and interpretability of the paper and will contribute to a more positive assessment in the final review. Below, we address each of these valuable suggestions that help improve the quality of the manuscript.
>
> ---
>
> > S1: Qualitative Comparison: While the quantitative comparison with FetalCLIP is justifiably omitted, the Related Work section would benefit from a deeper qualitative discussion. Positioning FetalMind vs. FetalCLIP within the clinical workflow and discussing their potential synergy would better contextualize your contribution.
>
> Thank you for your insightful comment. It is indeed helpful for contextualizing our contribution. By referring to step (1) **Image-to-View** in the clinical workflow shown in Fig. 1c of the manuscript, we provide a clearer explanation of the differences between the two methods in the Related Work section.
>
> Specifically, FetalCLIP mainly operates at the level of single-image interpretation within the clinical pipeline. It focuses on view classification, organ segmentation, and several related subtasks, and it is primarily used to assist clinicians in identifying standard views. Beyond this, FetalMind is the first to achieve holistic interpretation of fetal ultrasound images. It can directly generate complete clinical reports and diagnostic conclusions that support real clinical decision-making.
>
> We have added this discussion to the revised Related Work section and highlighted the changes in **Red**.
>
> ---
>
> > S2: Clarity on "SVPO": The name "SVPO" could be misleading. I suggest explicitly stating in the main text that the novelty lies in the "Salient View" data strategy for constructing preference pairs, not in a new "Preference Optimization" algorithm. This will prevent reader confusion.
>
> We thank the reviewer for pointing out the potential ambiguity in our naming. We agree that the term “SVPO” may misleadingly suggest that we are introducing a new preference-optimization algorithm.
>
> In the revised manuscript, in Section 4.3 *Salient Epistemic Disentanglement*, we have therefore renamed **Preference Optimization via SVPO** to **Data-Centric Learning via SVPO.** We also added the clarification: “The key idea is a data-centric strategy that constructs preference pairs by mining Salient Views from the knowledge graph on top of existing preference-optimization algorithms.” All modifications have been highlighted in **Red** in the updated version.
>
> ---
>
> > S3: Shortcut Learning Risk: The new real-world clinical evaluation is a strong addition. However, the theoretical risk of the model learning "splicing artifacts" from the synthetic data remains. Please briefly and candidly discuss this in the Limitations section to enhance the paper's rigor.
>
> Thank you for the valuable comment. We agree that synthetic data may introduce a theoretical risk of the model learning undesired **stitching artifacts**.
>
> In the revised version, we have added a discussion of this limitation in the *Limitations* section of the main paper and highlighted the changes in **Red**.

---

### Official Review · Reviewer_texo · 2025-10-27

**Soundness:** 3
**Presentation:** 3
**Contribution:** 3
**Rating:** 6
**Confidence:** 3

**Summary:**

This paper introduces FetalMind, a vision-language foundation model designed for fetal ultrasound report generation and diagnosis. The primary contributions are twofold: 1) The creation of FetalSigma-1M, a novel dataset of over 20,000 fetal ultrasound reports and 1.19 million images. 2) A new training methodology called Salient Epistemic Disentanglement (SED), which is guided by clinical workflows. Experiments show that FetalMind significantly outperforms a range of open-source and proprietary models on both diagnostic and report generation tasks.

**Strengths:**

1. This paper addresses the challenging task of fetal ultrasound interpretation, a domain that is not well-handled by current MLLMs.
2. This paper introduces FetalSigma-1M, a multi-center fetal ultrasound dataset.
3. This paper presents a thorough evaluation against a wide array of strong baselines with detailed analysis.

**Weaknesses:**

1. While the creation of FetalSigma-1M is a key strength, the paper indicates that only model weights, not the dataset, will be released. This limits the reproducibility of the work.
2. The method for evaluating baselines that lack native diagnostic capabilities involves using GPT to extract diagnoses from generated reports. This introduces a potential variable, as errors from the second step could unfairly penalize the baseline models.

**Questions:**

For the baseline models lacking native diagnostic capability, you employ GPT to perform diagnosis based on their generated reports.  Have you analyzed the potential error introduced by this two-step process? How can you ensure that the reported performance deficit of the baselines is not partly due to failures in this secondary diagnostic step? Is there a better way to evaluate these models?

---

> ### Author Response · Authors · 2025-11-18
> **Authors Rebuttal to Reviewer texo (1/2)**
>
> We sincerely thank you for your thorough feedback: the paper is clearly written, the proposed method is well-motivated, easy to reimplement, and the experiments are well organized. We address your questions below and have incorporated all feedback in the revised version. **All new material added to the revised manuscript has been highlighted in blue text for better visibility.**
>
>
> > 1. While the creation of FetalSigma-1M is a key strength, the paper indicates that only model weights, not the dataset, will be released. This limits the reproducibility of the work.
>
>
> A1: Valuable feedback! As noted by Review 1 (Vexq): *“The use of real clinical ultrasound data without a stated IRB approval or de-identification procedure may pose potential risks of patient privacy leakage.”*
>
> We would like to emphasize that we are committed to open science. To ensure reproducibility and verifiability of our experiments, we will release a subset of the dataset from three clinical centers as well as the model weights after the paper is accepted.
>
> In addition, our baseline model is built upon Qwenvl-2.5-7B, which offers strong flexibility and openness, making it highly suitable for domain-specific fine-tuning in medical applications.
>
> ---
>
> > 2. The method for evaluating baselines that lack native diagnostic capabilities involves using GPT to extract diagnoses from generated reports. This introduces a potential variable, as errors from the second step could unfairly penalize the baseline models.
>
> A2: Thank you for this excellent question. We have thoroughly verified this issue before drafting the manuscript. Our conclusion is that it does not introduce any significant impact. As shown in the table below, GPT-based diagnostic extraction can reproduce human-annotated diagnostic labels in the vast majority of cases, with an error rate within 1%. The reason the error remains so low is primarily due to two factors:
> - The semantic structure of radiology reports is highly stable;
> - The diagnostic extraction task is essentially a low-difficulty semantic matching problem for language models.
>
> **Analysis of GPT-Based vs. Direct Diagnosis. (*) indicates models fine-tuned with Supervised Fine-Tuning.**
>
> | Type | Model | #Params | GPT Diagnosis | P ↑ | R ↑ | F1 ↑ | ACC ↑ | Body F1-20 ↑ |
> |---|--|----|--|------|---|-------|---|------|
> | **w/ US Train** | Gemini 2.5 Pro | - | ✔ | 19.4 | 16.1 | 17.6 | 71.4 | 26.4 |
> | | GPT-5 | - | ✔ | 19.1 | 12.6 | 15.2 | 71.6 | 23.6 |
> | | InternVL3 * | 1B | ✘ | 26.2 | 18.9 | 22.0 | 78.2 | 39.9 |
> | | **FetalMind-S1** | **1B** | ✘ | 23.1 | **29.2** | 25.8 | 79.0 | 45.2 |
> | | **FetalMind-M7** | **7B** | ✘ | **34.7** | _28.2_ | **31.1** | **81.3** | **50.2** |
> | | _FetalMind-M7_ | _7B_ | ✔ | _34.2_ | 27.6 | _30.7_ | _80.6_ | _50.0_ |
>
> In the revised version, we added a new ablation experiment in **B.4 Analysis of GPT-Based vs. Direct Diagnosis**, demonstrating that our method can still achieve strong performance even when using GPT to extract diagnostic results from generated reports.
>
> ---
>
> > Q1. For the baseline models lacking native diagnostic capability, you employ GPT to perform diagnosis based on their generated reports. Have you analyzed the potential error introduced by this two-step process?
>
> A3: Thanks. Our work is inspired by the method of extracting diagnostic information from X-ray report generation, which inherently involves certain challenges. The sources of error may include the following three points:
>
> - GPT may misunderstand the semantics while parsing the report, such as overlooking negations or rhetorical expressions. For instance, " atrial septal defect " may be misinterpreted as " ventricular septal defect ". Our innovative design of Fetal Token injection aims to prevent the model from confusing such semantic distinctions by utilizing special keyword tokens.
>
> - Medical semantics often involve gradation terms (e.g., "mild", "suspicious"), and LLMs may misclassify these terms in different contexts.
>
> -	Different centers may use inconsistent phrasing for the same condition. For example, Center A’s report says: "Choroid plexus cyst noted in left ventricle", while Center B’s report says: "Anechoic lesion detected within the choroid plexus". Both describe choroid plexus cysts.
>
> These discussions have been added to **B.4 Analysis of GPT-Based vs. Direct Diagnosis** in the revised version.

---

> ### Author Response · Authors · 2025-11-18
> **Authors Rebuttal to Reviewer texo (2/2)**
>
> > Q2. How can you ensure that the reported performance deficit of the baselines is not partly due to failures in this secondary diagnostic step?
>
> A4: Thank you for the insightful comment! As shown in the table above, the performance improvement is not attributable to any bias introduced by the two-stage **report generation → GPT-based diagnosis** evaluation pipeline. In addition, the reports generated by our model demonstrate markedly higher overall quality, including structural consistency, clinical relevance, and linguistic accuracy, compared with all baseline systems. Consequently, the performance difference is reflected not only in diagnostic correctness but also in a broader improvement in report generation capability.
>
> ---
>
> > Q3. Is there a better way to evaluate these models?
>
> A5: We appreciate your comment! In our experiments and literature review, we found that simple keyword-matching approaches perform significantly worse than GPT-based extraction. Nevertheless, we agree that this direction is promising, and we have outlined a potential future solution: fine-tuning a dedicated fetal-ultrasound language model that maps reports to diagnoses for more reliable automatic interpretation.
>
> It is also worth noting that our curated list of diseases and their fine-grained descriptions, compiled from clinical textbooks, was primarily intended to enhance the diagnostic performance of baseline models that lack native diagnostic capability.

---

> ### Public Comment · ~He_Xiao1 · 2026-03-01
>
> ***Dear Reviewer texo,***
>
> I hope this message finds you well.
>
> I am one of the authors of Paper 1306, titled "Epistemic-Aware Vision–Language Foundation Model for Fetal Ultrasound Interpretation." We noticed that you did not participate in the post-rebuttal discussion, and we wanted to kindly ask whether your final assessment remains consistent with your initial positive review.
>
> Could you please confirm if your score still stands, or if you were considering raising it?
>
> Thank you very much for your time and consideration.
>
> Best regards,
>
> Xiao He, Huangxuan Zhao, Guojia Wan, Zhou Wei, Yanxing Liu, Juhua Liu, Yongchao Xu, Yong Luo, Dacheng Tao, Bo Du (Wuhan University & Nanyang Technological University)

---

### Official Review · Reviewer_Vexq · 2025-10-31

**Soundness:** 3
**Presentation:** 3
**Contribution:** 3
**Rating:** 6
**Confidence:** 4

**Summary:**

The paper introduces FetalMind, a multimodal vision–language foundation model for fetal ultrasound diagnosis. It aims to address the limitations of existing medical multimodal large models (e.g., LLaVA-Med, Med-Flamingo, BiomedGPT) that struggle with information redundancy and disease confusion in multi-view fetal imaging. The core idea is a Salient Epistemic Disentanglement (SED) mechanism combined with Salient View Preference Optimization (SVPO) and a bidirectional disease–view knowledge graph. Within a reinforcement learning framework, these components jointly refine the model’s multi-view reasoning to better align with real-world clinical workflows. In addition, the authors construct FetalSigma-1M, the first large-scale fetal ultrasound report dataset, covering 12 medical centers, 20,000 clinical reports, and over one million images. Experiments demonstrate that FetalMind surpasses GPT-5 and Gemini 2.5 Pro across both report generation and diagnosis tasks, achieving 98% accuracy in detecting nine major fetal abnormalities and an average +14% performance gain across different gestational stages.

**Strengths:**

1. The proposed SED mechanism integrates salient view selection and disease–view association learning, effectively disentangling key lesion information across multiple imaging views. It significantly reduces information collapse and disease confusion, offering strong innovation and interpretability from a clinical knowledge perspective.
2. The constructed dataset spans multiple gestational stages, medical centers, and over 300 disease categories, providing high representativeness and research value as a solid foundation for future fetal ultrasound AI studies.
3. FetalMind outperforms mainstream multimodal large language models (MLLMs), including GPT-5 and Gemini, in report generation, disease classification, and anomaly detection tasks. Detailed ablation studies further confirm the independent contributions of key modules such as Token Injection, Spatial Alignment, and SED.

**Weaknesses:**

1. Although the paper introduces a reinforcement learning–based optimization mechanism (SVPO), the core algorithmic derivation remains intuitive, lacking sufficient theoretical analysis or stability guarantees (e.g., convergence and optimality). In particular, the coupling between SED and SVPO is not formally explained.
2. While the model performs impressively on the FetalSigma-1M dataset, it lacks external multi-institutional validation or real prospective testing, leaving its clinical generalizability uncertain.
3. Although comparisons with DPO and GRPO are provided, the paper does not further discuss the design rationale of the reward signal or analyze how different reinforcement strategies affect stability and sample efficiency.

**Questions:**

1. When multiple pathologies coexist or imaging views contain noise, how does SED ensure that the model correctly identifies salient lesions rather than over-relying on biased views? Are there visualization results or quantitative metrics to support this claim?
2. Is the reward function in SVPO based on salient-view matching scores? If so, could the reward source introduce expert annotation bias? Is it possible to design an unsupervised or self-supervised variant?
3. How consistent is FetalMind’s performance across different ultrasound devices or manufacturers? Has any domain adaptation or feature standardization analysis been conducted?
4. Given the strong comprehension ability of general-purpose models such as GPT-5, how does FetalMind complement them? What is its potential for extension to multilingual or cross-modal clinical dialogue scenarios?
5. Can the model be seamlessly integrated into real clinical workflows? Does it incorporate privacy-preserving mechanisms (e.g., differential privacy) to enable deployment across multiple institutions?

**Details Of Ethics Concerns:**

The use of real clinical ultrasound data without a stated IRB approval or de-identification procedure may pose potential risks of patient privacy leakage.

---

> ### Author Response · Authors · 2025-11-18
> **Authors Rebuttal to Reviewer Vexq (1/4)**
>
> We sincerely thank you for your thorough feedback: the paper is clearly written, the proposed method is well-motivated, easy to reimplement, and the experiments are well organized. We address your questions below and have incorporated all feedback in the revised version. **All new material added to the revised manuscript has been highlighted in blue text for better visibility.**
>
> > 1. Although the paper introduces a reinforcement learning–based optimization mechanism (SVPO), the core algorithmic derivation remains intuitive, lacking sufficient theoretical analysis or stability guarantees (e.g., convergence and optimality). In particular, the coupling between SED and SVPO is not formally explained.
>
> A1: Thank you for your valuable feedback. We fully agree that providing stronger theoretical support for the proposed SVPO mechanism would further strengthen our work. Currently, the design of SVPO is primarily motivated by clinical observations and real-world requirements in fetal ultrasound tasks. We have included additional analysis in Section B.3, Analysis of Reinforcement Learning Methods.
>
> ### Relationship between SVPO and SED
>
> - **SVPO** can be generalized to multimodal large language models (MLLMs) that require multi-image reasoning. It can be seamlessly integrated into multi-image reasoning pipelines according to task-specific needs, enhancing both efficiency and accuracy by distinguishing salient images from non-salient ones.
>   *For example, when three images must be jointly analyzed but the key diagnostic information mainly lies in Image 1, SVPO effectively guides the model to focus on the most relevant visual evidence.*
>
> - **SED** introduces graph-based cognitive capabilities. It uses a bipartite graph structure to distinguish salient and non-salient images, and further incorporates SVPO to establish preference relations between abnormal images and diseases. This design enables disease–image graph relationships to be naturally injected into the MLLM’s reasoning process.
>   *This mechanism closely mirrors the workflow of clinicians: doctors identify key views, focus on abnormal regions across multiple images, and make integrated judgments based on disease knowledge.*
>   Therefore, SED is particularly suitable for multi-image reasoning tasks involving graph-structured relationships.
>
> In summary, **SED embeds and strengthens SVPO**: it not only models saliency among images but also leverages graph structures to support disease–image associative reasoning. We have added this analysis and discussion to Appendix B.2 Analysis of SVPO and SED in the revised version.
>
> ---
>
> > 2. While the model performs impressively on the FetalSigma-1M dataset, it lacks external multi-institutional validation or real prospective testing, leaving its clinical generalizability uncertain.
>
>
> A2: Good question! We have conducted multi-center external validation. As stated in line 319 of the main paper, we randomly split data from nine centers into training/validation/test sets with a 7:1:2 ratio. The FetalSigma-1M dataset includes 12 centers in total, among which 3 centers were reserved for multi-center external validation in our experiments.
>
> We believe that being the first to propose a multimodal large model for fetal ultrasound constitutes a sufficient contribution. To further verify the effectiveness of our method, we additionally performed real-world clinical scenarios testing on 56 cases from 2 clinical centers, as shown in Table 1. Specifically, we designed three groups: (1) a mid-level clinician, (2) a mid-level clinician assisted by our FetalMind model, and (3) a real clinical control group consisting of three clinicians including at least one senior clinician. After the image acquisition phase of the ultrasound examination, diagnostic interpretation was conducted in parallel.
>
> As shown in Table, FetalMind effectively assists clinicians by improving the completeness and consistency of report writing as well as enhancing diagnostic accuracy. We carefully collected and preprocessed the FetalSigma-1M dataset from real clinical scenarios rather than generic medical tasks, which enables FetalMind to acquire strong clinical applicability. In revised version, we have included this analysis in Appendix A.6 REAL-WORLD CLINICAL DECISION-MAKING ANALYSIS.
>
> **Overall comparison of NLP and classification metrics between Doctor and Doctor+AI**
>
> | Method     | BLEU-1  | BLEU-4  | ROUGE-1 | ROUGE-L | METEOR | Precision_micro | F1_micro |
> |----|---|----|--|---|----|---|---|
> | Doctor     | 75.388  | 67.817  | 77.450  | 72.019  | 30.592 | 0.568           | 0.562    |
> | Doctor+AI  | **88.532** | **81.605** | **86.002** | **85.717** | **59.351** | **0.679** | **0.653** |

---

> ### Author Response · Authors · 2025-11-18
> **Authors Rebuttal to Reviewer Vexq (2/4)**
>
> > 3. Although comparisons with DPO and GRPO are provided, the paper does not further discuss the design rationale of the reward signal or analyze how different reinforcement strategies affect stability and sample efficiency.
>
> A3: Thanks for your suggestion. DPO relies on a fixed reference policy $\pi_{\text{ref}}$ (typically the SFT model) and optimizes $L(\pi_\theta; \pi_{\text{ref}})$, which implicitly constrains $\pi_\theta$ to remain close to a reference that often underfits domain-specific medical knowledge. This may cause the model to overfit superficial stylistic cues in the preference data (e.g., verbosity, a confident tone), while compromising factual correctness.
>
> CPO removes dependence on $\pi_{\text{ref}}$ by adopting a uniform reference and adding a BC regularizer: ${E}_{(x,y_w)\sim D}.$ The BC term (i) prevents divergence from the true preferred data distribution and (ii) provides an absolute supervised signal on $y_w$, beyond relative “win–lose’’ comparisons. These properties are particularly important in medical applications, where behavior is highly deterministic and correctness is far more crucial than style. Empirically, reducing sampling temperature improves performance, consistent with CPO+BC shaping a sharper and more deterministic output distribution, which further justifies our choice of CPO with BC regularization over standard DPO.
>
> **GRPO performance degradation.** In our fetal ultrasound experiments, GRPO-based RL degrades performance relative to supervised training. The reward functions used by GRPO are only imperfect proxies for real clinical objectives and cannot fully capture fine-grained anatomical consistency, multi-view reasoning, or standardized reporting. The policy tends to overfit these proxy signals, “gaming’’ the reward through phrasing or templates, while degrading localization accuracy, measurement validity, and structural coverage.
>
> Additionally, GRPO introduces extra stochasticity and high-variance updates by optimizing relative rewards over sampled candidates, which disrupts the well-calibrated likelihood learned by supervised training in this near-deterministic task and leads to reward-seeking modes with reduced robustness. A promising direction is to train a dedicated reward model to provide more clinically aligned feedback for GRPO, especially for complex multi-image, multi-condition fetal ultrasound scenarios.
>
> In the revised version, we have added a full discussion of this component in Section B.3, *Analysis of Reinforcement Learning Methods*.
>
> **Analysis of Reinforcement Learning Methods**
>
> | Setting    | BLEU-4  | F1   | ACC  | AVG  |
> |------------|---------|------|------|------|
> | **Ours**   | **23.1** | **31.1** | **81.3** | **45.2** |
> | w/o SED    | 13.7    | 26.7 | 80.1 | 40.5 |
> | w/ GRPO    | 9.7     | 24.2 | 79.2 | 37.3 |
> | w/ DPO     | 7.9     | 12.3 | 65.8 | 28.7 |
> | Vanilla    | 9.2     | 25.8 | 79.0 | 38.0 |
>
> ---
>
>
>
> > Q1. When multiple pathologies coexist or imaging views contain noise, how does SED ensure that the model correctly identifies salient lesions rather than over-relying on biased views? Are there visualization results or quantitative metrics to support this claim?
>
> A4: As shown in Fig. 3c of the main paper, our model inherently supports the coexistence of multiple pathologies, where more pathologies naturally correspond to more salient views. Moreover, the SED mechanism aligns with clinically recognized salient views for each disease, rather than relying on arbitrary noise or atypical view patterns.
>
> To further demonstrate that disease recognition does not arise from view bias, we visualize the disease–view graph in Appendix A.7 *Visualization of the Disease–View Graph*. As illustrated, a single Four-Chamber Cardiac View can contain more than ten distinct fetal cardiac abnormalities, and the alignment between Diease–View relations is notably high, making it difficult to infer diseases solely from the combination of salient views. This evidence supports that our model does not rely on view bias to perform disease identification.

---

> ### Author Response · Authors · 2025-11-18
> **Authors Rebuttal to Reviewer Vexq (3/4)**
>
> >  Q2. Is the reward function in SVPO based on salient-view matching scores? If so, could the reward source introduce expert annotation bias? Is it possible to design an unsupervised or self-supervised variant?
>
> A5: We would like to clarify that the reward function in SVPO is derived from the CPO framework, where the reward signal is grounded in expert-annotated knowledge graphs. This supervision provides explicit positive and negative contrast, which is crucial for stable multi-image understanding. Without such contrastive structure, unsupervised or self-supervised variants may suffer from catastrophic failure, particularly due to the exponential increase in tokens across multiple images.
>
> It is possible to design a **self-supervised** variant of reward learning by leveraging the model’s own prediction consistency or multi-view contrastive learning mechanisms, allowing the model to generate pseudo reward signals from its internal feedback for self-optimization. On the other hand, **unsupervised** paradigms generally lack interpretability, especially in clinical applications. Therefore, we consider unsupervised approaches insufficient for addressing this problem.
>
> ---
>
> >  Q3. How consistent is FetalMind’s performance across different ultrasound devices or manufacturers? Has any domain adaptation or feature standardization analysis been conducted?
>
> A6: Thank you for your valuable feedback. FetalMind is trained on data collected from multiple ultrasound devices, including equipment from more than four different manufacturers. We have added this clarification to **A.1 Attention Analysis** in the revised version.
>
> **Domain Adaptation:**  Our proposed SED mechanism includes a design that specifically addresses cross-device sensitivity. In constructing sample pairs, the pathological images and the corresponding normal images may come from different ultrasound devices. Through this cross-device contrastive learning strategy, the model learns not only lesion-related salient features but also to explicitly suppress low-level variations associated with device characteristics, such as brightness, texture, artifacts, and probe-related patterns. In other words, SED treats semantic lesion consistency as the primary diagnostic alignment objective, while considering device-specific differences as noise factors that should be suppressed.
>
> **Feature Standardization Analysis:**  We also performed extensive standardization at the model input level. Specifically:
> - Sensitive information embedded in the raw images was standardized during preprocessing.
> - The Classify Block filters out low-quality images with weak diagnostic features while aligning view perspectives, thereby ensuring that only diagnostically meaningful images are retained.
>
> ---
>
> >  Q4. Given the strong comprehension ability of general-purpose models such as GPT-5, how does FetalMind complement them? What is its potential for extension to multilingual or cross-modal clinical dialogue scenarios?
>
> A7: Good idea! GPT-5 and FetalMind can complement each other to a certain extent in two main ways:
> - The outputs of FetalMind can be fed into GPT-5 to enhance the patient-friendliness and interpretability of medical reports and diagnoses.
> - For complex cases, the combination of FetalMind’s analytical results and other modal information can be provided to GPT-5 to assist clinicians in decision-making, potentially serving as a foundational tool for fetal ultrasound.
>
> In multilingual settings, GPT-5 can automatically convert FetalMind’s structured diagnostic information, lesion descriptions, and imaging conclusions into natural expressions in different languages, enabling both patients and medical professionals to understand the diagnosis in their native language. However, for rigorous clinical applications, introducing general-purpose models such as GPT-5 may impact workflow efficiency. For example, fetal ultrasound reports typically follow a relatively structured format and do not rely heavily on extensive natural language descriptions. Directly employing large language models for report generation may introduce unnecessary verbosity, semantic inconsistencies, or longer generation time, thereby prolonging the clinical decision-making process.

---

> ### Author Response · Authors · 2025-11-18
> **Authors Rebuttal to Reviewer Vexq (4/4)**
>
> >  Q5. Can the model be seamlessly integrated into real clinical workflows? Does it incorporate privacy-preserving mechanisms (e.g., differential privacy) to enable deployment across multiple institutions?
>
> A8: Of course, as demonstrated in the real-world clinical scenarios study, we have seamlessly integrated the model into real clinical workflows. The workflow proceeds as follows: *begin acquisition → collect fetal ultrasound images → finish acquisition (AI analysis begins) → the clinician organizes equipment and assists the patient → the clinician reviews the collected ultrasound images and the AI-generated analysis to produce the final report and diagnosis.*
>
> Since the AI report generation takes only about 30 seconds, we observed in real deployment that this duration is significantly shorter than the time required for organizing equipment and assisting the patient. This includes cleaning the acoustic coupling gel from the patient’s skin, ensuring safe repositioning, and completing bedside procedures. Therefore, the AI component does not slow down or interfere with the clinical workflow. On the other hand, we thoroughly removed all potentially sensitive information during the reporting and data preprocessing stages to eliminate any risk of privacy leakage. The output of the FetalMind model contains only anonymized diagnostic results and lesion characteristics, without any personally identifiable information.

---

> ### Public Comment · ~He_Xiao1 · 2026-03-01
>
> ***Dear Reviewer Vexq,***
>
> I hope this message finds you well.
>
> I am one of the authors of Paper 1306, titled "Epistemic-Aware Vision–Language Foundation Model for Fetal Ultrasound Interpretation." We noticed that you did not participate in the post-rebuttal discussion, and we wanted to kindly ask whether your final assessment remains consistent with your initial positive review.
>
> Could you please confirm if your score still stands, or if you were considering raising it?
>
> Thank you very much for your time and consideration.
>
> Best regards,
>
> Xiao He, Huangxuan Zhao, Guojia Wan, Zhou Wei, Yanxing Liu, Juhua Liu, Yongchao Xu, Yong Luo, Dacheng Tao, Bo Du (Wuhan University & Nanyang Technological University)

---

### Author Response · Authors · 2025-11-18
**Summary of the Authors Response**

Dear Reviewers,

Thank you for taking the time to review our manuscript and for providing thoughtful and constructive feedback. We are delighted to see that the reviewer acknowledged the paper is *strong innovation and interpretability*, *high representativeness and research, solid foundation*  (Vexq),  *thorough evaluation strong, baselines* (texo), *enduring value for the community* (tEu6), *help the community further advance* (bzL1). Below we summarize some major points we have addressed in the rebuttal. More detailed responses are provided individually for each reviewer.

**Prospective Study.** We have additionally included a real-world clinical scenarios study, which provides real-world clinical evidence supporting the robustness and practical utility of our method. This real-world clinical evaluation demonstrates that the model can be integrated into routine clinical workflows. (Reviewer tEu6, bzL1)

**Details of the disease–view bipartite graph.** We have added both a Sankey visualization and a more detailed description of the disease–view bipartite graph. The enhanced explanation helps clarify the diagnostic relevance of these connections and how they contribute to the overall interpretability of our model. (Reviewer tEu6, bzL1)

**Reinforcement Learning Analysis.** To address the reviewer's concerns, we have expanded our discussion of CPO, GRPO, and DPO, outlining their respective advantages and limitations and clarifying why CPO with BC regularization provides more stable and clinically reliable behavior in our setting. (Reviewer Vexq, tEu6)

**Impact of GPT’s diagnostic outputs.** We have also added results comparing FetalMind’s direct diagnoses with those derived from GPT-generated reports, showing only minimal differences and demonstrating consistent diagnostic reliability across both modes. (Reviewer texo)

**Practicality of swapping views.** We have added a discussion of multi-lesion cases and performed a related analysis of view swapping. In addition, we have validated the effectiveness of view swapping from a clinical perspective. (Reviewer tEu6,bzL1)
In the point-to-point responses below, we have provided more detailed responses to each of the reviewers' concerns. **All new material added to the revised manuscript has been highlighted in red text for better visibility.** Please don't hesitate to let us know if there are any additional clarifications or experiments that we can offer!

Yours Sincerely,

Authors

---

### Author Response · Authors · 2025-12-01
**Author Summary of Revisions and Clarifications**

Dear PCs, ACs, and Reviewers,

We are truly grateful for the invaluable time and detailed feedback provided by all the reviewers.

This year, the ICLR community has made significant efforts to improve review quality. We sincerely thank the ACs for sending reminder messages to reviewers who had not posted their responses, and we also thank the PCs for extending the discussion period. Several authors of this paper also served as reviewers; we diligently read rebuttals from other papers, ensuring that concerns were addressed and asking follow-up questions as needed.

In this comprehensive response, we summarize the two major revisions incorporated into the new manuscript based on the reviewers’ valuable suggestions, aiming to address any remaining concerns from the reviewers, AC, and SPC.

---

- **Prospective Study.** We have additionally included a real-world clinical scenarios study, which provides real-world clinical evidence supporting the robustness and practical utility of our method. This real-world clinical evaluation demonstrates that the model can be integrated into routine clinical workflows.

- **Reinforcement Learning Analysis.** To address the reviewer's concerns, we have expanded our discussion of CPO, GRPO, and DPO, outlining their respective advantages and limitations and clarifying why CPO with BC regularization provides more stable and clinically reliable behavior in our setting.

---

**Clarification regarding Reviewer tEu6’s comments**

Reviewer **tEu6** initially `misunderstood` the comparison methods when providing the `initial score (4)`. After our `clarification`, the reviewer acknowledged that **“the quantitative comparison with FetalCLIP is justifiably omitted”**, further noting that **the new experiments and clarifications have resolved most of my initial concerns and significantly improved the paper** and attempted to raise the score.

---

During the discussion phase, we are committed to engaging all reviewers in the conversation, regardless of their level of familiarity with the field, and we appreciate the **AC’s** efforts in promoting a fair and constructive exchange.

Best regards,

All Authors

---

### Note · Authors · 2026-04-10

**Comment:**

Due to the identity leakage incident at ICLR 2026, which fundamentally violated the double-blind review principle, we regretfully withdraw this paper and decline to further discuss it with reviewers.

**Withdrawal Confirmation:**

I have read and agree with the venue's withdrawal policy on behalf of myself and my co-authors.

---

### Meta-Review · Area_Chair_BFQ7 · 2026-01-21

**Summary:**

Four reviewers have evaluated this work and initially scored it as 6, 6, 4 and 6. The paper proposes a system for fetal
ultrasound report generation and diagnosis. The system handles a variable number of views and tries to mimic how clinicians would look over them by disentangling salient and normal views.

This paper truly is a borderline paper. Among positives is the practical design which takes fetal ultrasound diagnosis specifically under consideration. On negative side, reviewers note that algorithmic innovation lacks sufficient theoretical analysis and the coupling between SED and SVPO could be formalized. Another key concerns is that only model weights, not the dataset, would be released which undermines one of key contributions. Finally, although practical, the paper is somewhat incremental in that SVPO 's core loss is identical to that of CPO, and th augmentation strategy is not bullet proof.

On balance, the paper is not ready for publication. A major advantage would be the full data release - authors mention a possibility of releasing a subset but the details of what is possible remain vague at this stage.

**Reviewer Concerns:**

Comparison to Reinforcement Learning Methods.

Comparison of GPT-Based vs. Direct Diagnosis.

Real world scenario comparisons (at least partially).

**Reviewer Scores:**

Reviewer Vexq would likely decrease or maintain score 6 due to concerns about SVPO and lack of further theoretical advancements.

Rviewer texo would likely decrease the score due to the lack of possibility to release the entire dataset (reproducibility issue).

Reviewer tEu6 was concerned about augmentations and possible "anatomically implausible combinations". The argument about empirical gains might have not been sufficient for this reviewer to increase the score from 4.


Reviewer bzL's concerns about efforts needed to "create and verify the bipartite knowledge graph" would likely remain not fully addressed.

---

> ### Public Comment · ~He_Xiao1 · 2026-03-01
> **Question About Meta-Review (Paper 1306)**
>
> ***Dear Dr. Area Chair BFQ7,***
>
> In your Meta-Review for Paper 1306, you stated that three reviewers who gave positive scores but did not join the discussion would "likely decrease the score."
>
> Since these reviewers remained silent for the entire three-week discussion period, could you please explain why you believe they intended to lower their scores?
>
> Thank you for your clarification.
>
> Best regards,
>
> Xiao He, Huangxuan Zhao, Guojia Wan, Zhou Wei, Yanxing Liu, Juhua Liu, Yongchao Xu, Yong Luo, Dacheng Tao, Bo Du (Wuhan University & Nanyang Technological University)

---

### Decision · Program_Chairs · 2026-01-26

Reject

---

> ### Public Comment · ~He_Xiao1 · 2026-03-01
> **Formal Complaint Regarding ICLR AC Conduct (Paper ID: 1306)**
>
> ***Dear Dr. Program Chairs,***
>
> We understand this has been an exceptionally demanding year for the ICLR team, and we sincerely appreciate all the hard work that goes into coordinating the review process for such a large-scale conference.
>
> **We are writing to respond to the urgent request for ethics clarification. However, we must simultaneously lodge a formal appeal regarding the handling of Paper 1306, "Epistemic-Aware Vision–Language Foundation Model for Fetal Ultrasound Interpretation." The AC BFQ7 has seriously violated ICLR guidelines by repeatedly threatening, without any justification, that our reviews would "likely decrease the score."**
>
> ***Specifically, regarding a comparable paper, FETAL-GAUGE (Paper ID 23959), which was accepted: unlike our submission, this work has neither open-sourced its dataset nor demonstrated corresponding algorithmic innovation, practical verification to date and Theoretical Analysis. We believe the Area Chair BFQ7 has seriously violated ICLR guidelines.***
>
> We believe there are serious procedural irregularities involving Area Chair BFQ7, which threaten the integrity of the review process for our submission.
>
> **Formal Complaint: Unfounded Speculation & Disregard for Evidence**
>
> A. Unfounded Speculation on Scores (Violation of Fairness) In the Meta Review, Area Chair BFQ7 explicitly stated that three reviewers who had given positive evaluations but did not respond to the rebuttal would "likely decrease the score." This statement stands in direct violation of the ICLR Code of Ethics and Area Chair Guidelines on the following counts:
>
> Imputed Opinions: The AC is interpreting reviewer silence as an intent to lower scores. This is a baseless assumption. Standard academic protocol dictates that if a reviewer does not engage post-rebuttal, their original scores and comments stand as the official record.
>
> Lack of Evidence: There is no record in the discussion log to support the claim that these reviewers intended to lower their scores. The decision is based purely on the AC's subjective projection rather than the actual consensus.
>
> Procedural Unfairness: Penalizing authors for the reviewers' lack of response by assuming a negative outcome is procedurally unjust.
>
> Related PAPER: Regarding a comparable paper, FETAL-GAUGE (Paper ID 23959), which was accepted: unlike our submission, this work has neither open-sourced its dataset nor demonstrated corresponding algorithmic innovation, practical verification to date and Theoretical Analysis. All of the aforementioned points could serve as valid grounds for rejection by AC BFQ7. Therefore, We believe the Area Chair BFQ7 has seriously violated ICLR guidelines.
>
> B. Negligence Regarding Submitted Materials Furthermore, the current "urgent" request for ethics clarification demonstrates a lack of attention to our file. We already submitted these exact materials 10 days ago, strictly following the previous instructions. The fact that we are being asked to provide the same information again suggests that the AC has ignored our previous communication, just as they have ignored the actual recorded scores of the reviewers.
>
> We respectfully request that the Program Chairs review the AC's handling of Paper 1306 to ensure an evidence-based and fair assessment.
>
> Sincerely,
>
> ***Xiao He, Huangxuan Zhao, Guojia Wan, Zhou Wei, Yanxing Liu, Juhua Liu, Yongchao Xu, Yong Luo, Dacheng Tao, Bo Du (Wuhan University & Nanyang Technological University)***